# *Articulation in Motion*: Prior-free Part Mobility Analysis for Articulated Objects By Dynamic-Static Disentanglement

**Hao Ai**[*,1], **Wenjie Chang**[*,2], **Jianbo Jiao**[★,1], **Ales Leonardis**[★,1], **Eyal Ofek**[†,*,1]

[1] School of Computer Science, University of Birmingham, Birmingham, UK
[2] University of Science and Technology of China

## Abstract

Articulated objects are ubiquitous in daily life. Our goal is to achieve a high-quality reconstruction, segmentation of independent moving parts, and analysis of articulation. Recent methods analyse two different articulation states and perform per-point part segmentation, optimising per-part articulation using cross-state correspondences, given a priori knowledge of the number of parts. Such assumptions greatly limit their applications and performance. Their robustness is reduced when objects cannot be clearly visible in both states. To address these issues, in this paper, we present a new framework, *Articulation in Motion (*AiM*)*. We infer part-level decomposition, articulation kinematics, and reconstruct an interactive 3D digital replica from a user–object interaction video and a start-state scan. We propose a dual-Gaussian scene representation that is learned from an initial 3DGS scan of the object and a video that shows the movement of separate parts. It uses motion cues to segment the object into parts and assign articulation joints. Subsequently, a robust, sequential RANSAC is employed to achieve part mobility analysis *without any part-level structural priors*, which clusters moving primitives into rigid parts and estimates kinematics while automatically determining the number of parts. The proposed approach separates the object into parts, each represented as a 3D Gaussian set, enabling high-quality rendering. Our approach yields higher quality part segmentation than previous methods, without prior knowledge. Extensive experimental analysis on both simple and complex objects validates the effectiveness and strong generalisation ability of our approach. Project page: https://haoai-1997.github.io/AiM/.

> "Motion is the cause of all life."
>
> *Leonardo da Vinci*

## 1 Introduction

Everyday environments are abounded with articulated objects[*], composed of multiple rigid parts linked by joints (Mueller, 2019) (*e.g.* doors with revolute joints and drawers with prismatic joints). Modelling of such objects is valuable for practical applications across scene understanding (Jia et al., 2024; Huang et al., 2024b), robotics (Kerr et al., 2024; Wu et al., 2025b), mixed reality (MR) (Taylor et al., 2020; Jiang et al., 2024), and embodied AI applications (Puig et al., 2023; Zhou et al., 2025). Advances in neural 3D representations (Mildenhall et al., 2021; Kerbl et al., 2023; MacSwayne et al., 2025) enable high-fidelity object-level 3D reconstruction; however, reconstructing part-level structure, articulation dynamics, and functionality of articulated objects remains challenging.

Substantial efforts have been devoted to building 3D physics-consistent and interaction-ready assertions of articulated objects from RGB or RGB-D observations (Wei et al., 2022; Song et al., 2024;

---

[*]Equal contribution. [†]Correspondence: e.ofek@bham.ac.uk. [★]Equal advice.
[*]In this work, we only discuss the human-made articulated objects with rigid parts.

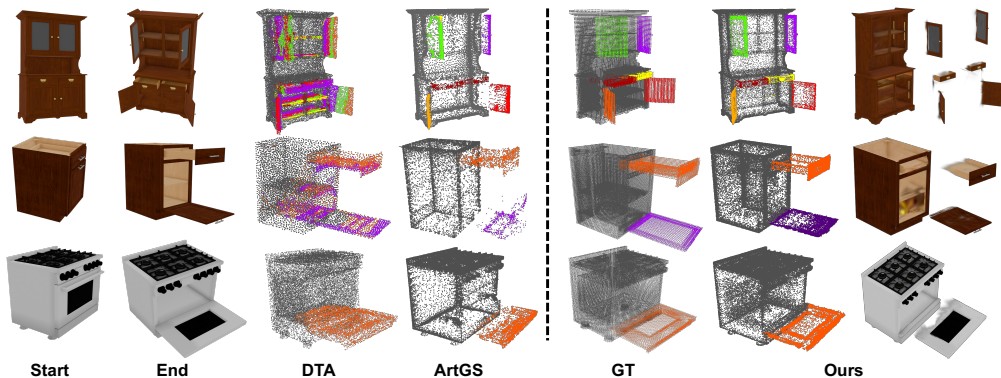

Figure 1: **Left**: Prior two-state methods often degrade on the sequences from closed-start to open-end. **Right**: Results of the proposed AIM, compared to ground truth (GT) geometry.

Wu et al., 2025a). Some approaches (Mu et al., 2021; Jiang et al., 2022b; Heppert et al., 2023) rely on the parameters of known joints in learning articulated shape representations. However, collecting such data at scale can be time-consuming and often has limited generalisation to previously unseen objects. To mitigate these issues, some unsupervised, part-level reconstruction methods have been proposed (Liu et al., 2023a; Deng et al., 2024; Weng et al., 2024; Liu et al., 2025). Most of them assume multi-view observations of the objects in two distinct articulation states, denoted as *start* and *end* states. Liu et al. (2023a); Deng et al. (2024) recovered a deformation field between the NeRF-based start-state and end-state geometries. However, optimisation is unstable and highly sensitive to initialisation. Similarly, the 3D Gaussian splatting (3DGS) based method (Wu et al., 2025a) learns a per-Gaussian start-to-end deformation field, enabling static-dynamic segmentation of up to one moving part per step. As deformation is defined over all Gaussians, threshold-based separation is prone to noise. Lately, given a known number of articulated parts, DTA (Weng et al., 2024) simultaneously reconstructs a *start* and *end* point cloud, predicts per-part segmentation probabilities, and estimates articulation parameters via linear blend skinning (Kavan et al., 2007) to align the parts across the two states. Meanwhile, ArtGS (Liu et al., 2025) constructs *start* and *end* Gaussian sets and uses their geometric correspondence to initialise a canonical mid-state Gaussian set. It then learns part-centre locations, predicts Gaussian-to-part assignments (Huang et al., 2024c), and optimises per-part articulation following the blend skinning (Song et al., 2024). Although effective, these two-state methods degrade substantially when the number of articulated parts is unknown (as shown in Fig. 2), and their stability is limited by the reliance on geometric correspondence between the two states. Specifically, the commonly used two-state input setting has inherent limitations: *Many articulated objects cannot be well represented by only a start and an end state*. When the end state reveals regions absent in the start state, breaking cross-state correspondence (*e.g.* the interior of a refrigerator or oven, as shown in Fig. 2), these methods are prone to degraded segmentation.

In this paper, we introduce *Articulation in Motion (*AIM*)*, a new framework that reconstructs the geometry, segmentation, and kinematics of articulated objects by analysing a video of their articulated motion, which is simple, practical, and better aligned with the way humans learn articulation through continuous interaction (Fig. 3) rather than using isolated start and end states. Furthermore, continuous motion cues avoid cross-state correspondence failures when the end state reveals newly seen regions (see Fig. 1 and Fig. 2). *We do not assume a known number of articulated parts, any prior knowledge of their joint types or motion parameters, or visibility along the entire motion.* We recover the articulation parameters stably for interactive manipulation. It comprises three stages (see Fig. 3 and Fig. 5): *I)* 3DGS is used to reconstruct the initial geometry and appearance. *II)* We introduce a dual-Gaussian scene representation, which contains the pre-built start-state Gaussian and a deformable 3DGS, which tracks motion on the interaction video. A pre-built start-state Gaussian is gradually pruned as a static base, achieving dynamic-static disentanglement based on the motion cues. *III)* Depending on the trajectories of only-moving Gaussians, an optimisation-free sequential Random Sample Consensus (RANSAC) clusters them into rigid parts and estimates per-part articulation parameters *without any part prior*. Our contributions are summarised as follows:

- We present *Articulation in Motion (*AIM*)*, which reconstructs part-level articulated objects, with extracted joints, based on a video that shows the objects' degrees of freedom. It opens a way to use interactive natural videos for reconstruction.

- We propose a *dual-Gaussian representation* to disentangle the statics and dynamics, and track the moving primitives. Additionally, we introduce a static-during-motion detection module to handle newly revealed but static regions during interaction.
- Our method achieves robust part segmentation and articulation estimation using Sequential RANSAC, without any structural prior.
- Extensive experiments demonstrate that our method can independently segment stably and accurately moving parts of the object, reconstruct the geometry and articulation parameters of each part, and its appearance, under challenging scenarios.

## 2 RELATED WORKS

**Articulated object reconstruction from videos or images.** This work focuses on articulated objects, composed of rigid parts and connected by joints. For such objects, research has focused primarily on improving piecewise rigidity, identifying part-level mobility, and enabling controllable 3D model generation. REACTO (Song et al., 2024) reconstructs the canonical object state, represented by NeRF, and learns a deformation field with enhanced part rigidity from a captured video. However, REACTO reconstructs articulated objects as a single unified surface, without part-level geometry, which limits physically realistic interaction. Jiang et al. (2022b); Liu et al. (2023a); Deng et al. (2024); Weng et al. (2024); Liu et al. (2025) proposed to reconstruct articulated objects at the part level and estimate joint parameters from multi-view RGB/RGB-D observations of two different articulation states, *i.e.* the state before interaction and the end state after interaction. PARIS (Liu et al., 2023a) learns a deformable field that applies two inverse motion parameters to a hypothetical intermediate-state NeRF. Similarly, REArtGS (Wu et al., 2025a) builds on 3DGS to learn a static-to-dynamic deformation field for the intermediate state and identify the dynamic part. Both are limited to objects with one moving sub-part. To support multiple movable parts, Weng et al. (2024); Deng et al. (2024); Liu et al. (2025) directly predict partwise segmentation probabilities for each point and learn the motion parameters per part to construct the cross-state correspondence fields, similar to linear blend skinning (Kavan et al., 2007).

**Part mobility analysis.** Part mobility analysis typically involves part segmentation and articulation estimation, *e.g.* joint type, axis and state. For supervised learning, Jiang et al. (2022a); Qian et al. (2022); Sun et al. (2024); Wang et al. (2024) leverage advanced network architectures to predict part mobility directly from a single RGB image or a motion video, while Weng et al. (2021); Liu et al. (2022; 2023b) jointly predict part-level 3D segmentation and per-part motion properties from a single point cloud. To reduce the dependence on annotated datasets, recent work has explored unsupervised solutions. A representative two-state-based pipeline (Zhong et al., 2023; Liu et al., 2023a; Weng et al., 2024; Wu

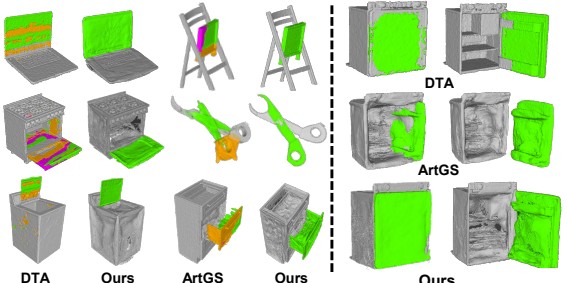

Figure 2: **Left:** DTA and ArtGS fail to recover from an incorrect input number of parts (4 here) and result in oversegmentations; **Right**: Visual results of DTA and ArtGS with closed-start and open-end states. The static part is gray and the moving part is green. In contrast, Ours requires no geometric priors and recovers accurate part-level segmentation from the continuous closed-start→open-end interaction process.

et al., 2025a) predict the part segmentation probabilities for each point and the articulation parameters per part, supervised by the point correspondence field between 3D shapes of two given input states. However, these methods depend on the input part number, lacking sufficient generalisation to real-world scenarios with unknown structural details, *e.g.* for objects with unknown structural details, optimisation becomes unstable. It often fails to converge to the correct number of parts (see Fig. 2). Additionally, as shown in Fig. 2, capturing multi-view observations for two distinct states can easily introduce ambiguities when cross-state correspondences are undefined for regions that appear only in the end state, instead, inspired by (Shi et al., 2021; Yan et al., 2019), which segment point clouds using trajectories from registered sequences, our framework infers part-level structure and articulation information from the motion trajectories in a single video. RSRD (Kerr et al., 2024), POD (Wu et al., 2025c) and Video2Articulation Peng et al. (2025), use a similar video-based input,

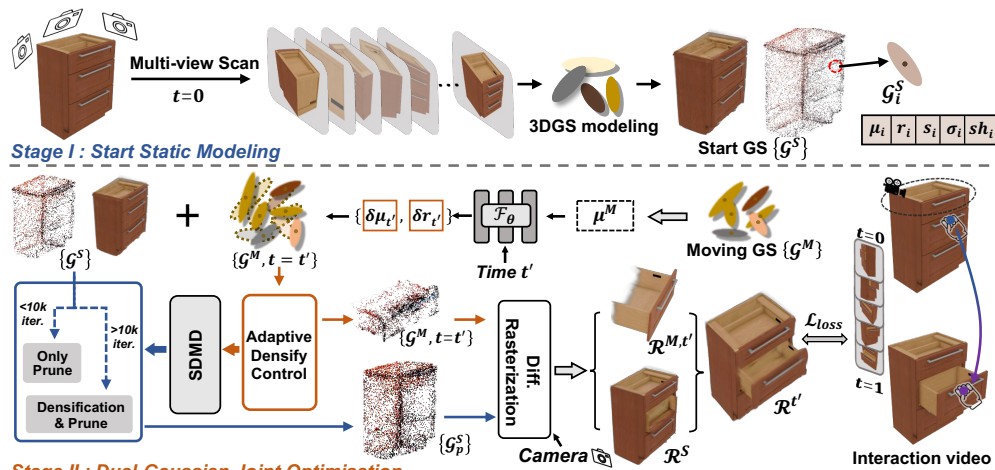

Figure 3: **Overview of the first two stages:** I) 3DGS start-state $\{\mathcal{G}^S\}$ reconstruction from a multi-view RGB scan. II) A deformable 3DGS $\{\mathcal{G}^M, t\}$ tracks motion video, while joint optimisation prunes moving components from $\{\mathcal{G}^S\}$. Pruned static Gaussian set $\{\mathcal{G}_p^S\}$ encodes the static base. An SDMD module handles newly revealed but static Gaussians. Together, these yield two separated Gaussian sets ($\{\mathcal{G}_p^S\}$ and $\{\mathcal{G}^M, t\}$) for the articulation analysis (Fig. 5).

but both focus on per-part pose tracking and require segmentation masks from pre-trained models. They cannot perform part segmentation autonomously, and the performance is fundamentally bounded by the pre-trained segmentation models (see Fig. A7).

**Dynamic Gaussian splatting.** 3DGS (Kerbl et al., 2023) provides an explicit point-based representation, enabling real-time, differentiable splatting-based rendering. As a result, there is increasing interest in extending 3DGS to dynamic scene modelling (Duisterhof et al., 2023; Luiten et al., 2024; Yang et al., 2024; Wu et al., 2024). Luiten et al. (2024) tracks attribute changes of each Gaussian primitive while Yang et al. (2024) learns an MLP-based deformation field from time and primitive positions to represent scene flow. Additionally, several methods (Wu et al., 2024; Duisterhof et al., 2023) introduce more efficient representations to encode temporal and structural information, and employ motion clustering strategies for compactness. Tracking all Gaussians is expensive, and motion can reduce motion-based segmentation (see Fig. A14 in Appendix). Our dual-Gaussian detects static parts of the geometry, represented by 3DGS, while moving geometry is tracked using deformable 3DGS; clear dynamic-static disentanglement enables stable segmentation.

## 3 OUR METHOD

The proposed *Articulation in Motion* includes three stages. *Stage I:* We reconstruct a 3DGS model (preliminaries *e.g.* 3DGS and Deformable 3DGS please see Appendix A.) of the object on an initial static state $\{\mathcal{G}^S\}$. *Stage II:* Given a video in which parts of the object are moved, we propose a dual-Gaussian representation (Sec. 3.1) that jointly optimises $\{\mathcal{G}^S\}$ that models the static part of the object and a deformable GS $\{\mathcal{G}^M, t\}$ that captures the moving parts of the object. Moreover, an additional static-during-motion detection (SDMD) module handles the newly static parts that are revealed during the video and adds them to the static part of the object. After obtaining $\{\mathcal{G}^M, t\}$ with the time-dependent deformation, we infer the trajectories of each moving primitive and introduce the sequential RANSAC to group the moving primitives in *Stage III*, achieve motion-based part segmentation and articulation estimation (Sec. 3.2). The entire training is supervised solely by RGB observations: the start-state scan and the monocular video frames. Below we describe AIM in detail:

### 3.1 DUAL-GAUSSIAN FOR DYNAMIC-STATIC DISENTANGLEMENT

To achieve motion-based part segmentation and articulation analysis, it is essential to accurately describe the trajectories of Gaussian primitives based on the given motion video. Although D-3DGS (Yang et al., 2024) can learn time-dependent deformation fields from motion cues in the video, it assigns a displacement to all Gaussians, including static ones. This introduces noise that may harm segmentation and articulation estimation, especially with multiple moving parts, where

all-Gaussian trajectories confuse the part-level structure (see Fig. A14 in Appendix). To address this issue, we propose a dual-Gaussian representation that comprises two sets of Gaussians to separately model the static base body and the moving components in the given video. The methodology is visually summarised in Fig. 3. Firstly, we train a vanilla 3DGS based on multiple views of the object in a start state, namely $\{\mathcal{G}^S\}$, following Eq. 8. Subsequently, given the motion video, we follow D-3DGS to initialise a moving Gaussian set $\{\mathcal{G}^M, t\}$ and train it with Eq. 9, to make these Gaussians capture the moving parts in the video and predict the spatiotemporal deformation field. We jointly render and optimize both sets, $\{G^S\} \cup \{G^M, t\}$, directing $\{\mathcal{G}_t^M\}$ to model motion cues and removing these moving elements from $\{\mathcal{G}^S\}$ to obtain the static base $\{\mathcal{G}_p^S\}$, achieving clean dynamic–static disentanglement for subsequent part mobility analysis. Newly emerging regions, initially captured by the moving Gaussian set, are identified by a static-during-motion detection (SDMD) module, which locates locally rigid components, estimates their local rigid motions, and reassigns them to the static set according to the predicted transformations.

**Dual-Gaussian joint optimisation.** Using the multi-view scan of the start state, we model the object's geometry and appearance via the original 3DGS pipeline. Following the standard training settings, we obtain a set of initial Gaussians $\mathcal{G}^S(\mu_i, s_i, r_i, \sigma_i, h_i)_{i=1}^{N_s}$, where $N_s$ denotes the total number of Gaussians, including both static and dynamic components. Thereafter, we initialise a sparse point cloud and prepare to learn a time-indexed deformable Gaussian set, denoted as $\{\mathcal{G}^M(\mu_j, s_j, r_j), t\}_j^{N_m}$. Then, we employ an MLP-based deformation network $F_\theta$ alongside the moving Gaussian set to capture the motion trajectory. Specifically, $\{\mathcal{G}^M(\mu_j, s_j, r_j)\}_j^{N_m}$ represents the geometric priors in the canonical space, while the changes $(\delta\boldsymbol{\mu}, \delta\boldsymbol{r})$ in the position and rotations are learned by the deformation network as:

$$(\delta\mu_j, \delta r_j) = \mathcal{F}_\theta(\gamma(sg(\mu_j)), \gamma(t)), \tag{1}$$

where $t$ is the input time index, $sg$ represents a stop-gradient operation, and $\gamma$ indicates the position encoding (Vaswani et al., 2017). We employ the same network architecture as D-3DGS Yang et al. (2024). To constrain the moving Gaussian set to encode only continuously moving content in the video, while the start-state Gaussian set $\{\mathcal{G}^S\}$ remains static-focused, we jointly optimise these two Gaussian sets. As shown in Fig. 3, during the initial 10k iterations of the optimisation, we freeze all attributes of $\{G^S\}$ except opacity $\sigma$, while $\{\mathcal{G}^M, t\}$ and the deformation network $\mathcal{F}_\theta$ are trained with the normal adaptive density control.

In this process, we progressively prune the moving elements of $\{\mathcal{G}^S\}$ as their opacity decreases over time to obtain the static Gaussian set, namely $\{\mathcal{G}_p^S\}$, while $\{\mathcal{G}^M, t\}$ fits the moving components in the video (see Fig. A4). In the following iterations, we unfreeze the static Gaussian set, and jointly perform densification and pruning on both sets. Through the differentiable rendering on the combination of $\{\mathcal{G}_p^S\}$ and $\{\mathcal{G}^M, t=t'\}$, we supervise the total training process with the video frame at the corresponding timestep $t=t'$. Since previously unseen areas in the start state, *e.g.* the interior structures of refrigerators, washing machines, and cabinets, will be captured by the moving Gaussian set, an SDMD detection module is introduced to audit the moving Gaussian set and prevent static leakage.

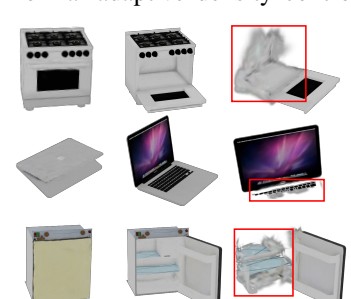

Figure 4: Renderings of the start (left), and end (middle) states. Without SDMD detection, some newly revealed static parts are wrongly associated with the moving Gaussian set (right).

**Static-during-motion detection.** During the first 10k iterations, we freeze all position-related attributes of $\{\mathcal{G}^S\}$, allowing $\{\mathcal{G}^M, t\}$ to thoroughly learn the moving parts while also adapting to newly revealed static content (Fig. 4). Although such content becomes stationary once revealed, it is often already occupied by moving Gaussians, which hampers the static set from learning this geometry. To address this, we introduce a static-during-motion detection (SDMD) scheme. During joint densification and pruning, we perform trajectory inference for the moving Gaussians $\{\mathcal{G}^M, t\}$ every 2,000 iterations at $t \in \{0, 0.5, 1\}$. We then apply sequential RANSAC with the Kabsch algorithm (Magri & Fusiello, 2016) and a fixed inlier threshold of $0.05$ to the resulting trajectory sequence to extract locally rigid motion patterns (details in Sec. 3.2). Groups whose motion magnitude falls below the preset threshold (defined in Sec. 3.2) are identified as static, and their Gaussian primitives are reassigned from $\{\mathcal{G}^M, t\}$ to $\{\mathcal{G}_p^S\}$. Compared with a simple motion-distance filter, our SDMD avoids misassignment near the joint axis (where motion trajectories are near zero).

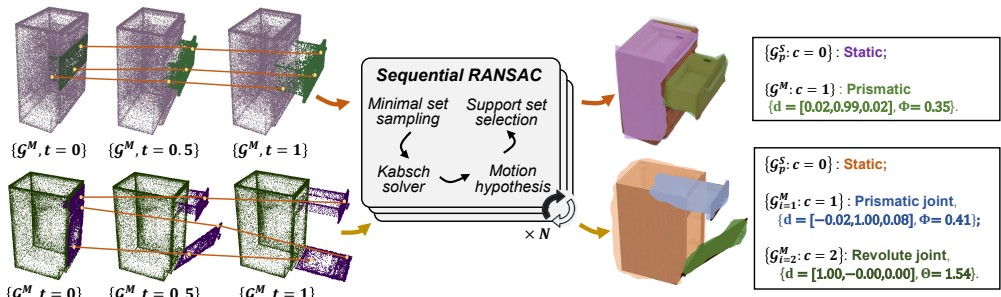

Figure 5: **Stage III**: **Motion-based part segmentation and articulation analysis**. As the clean $\{\mathcal{G}^M, t\}$ provides time-varying trajectories, Sequential RANSAC groups trajectories into rigid parts (multi-part supported) without priors or optimisation, and directly outputs per-part articulation parameters. The green (top) and purple (bottom) points are our predicted moving Gaussians.

## 3.2 MOTION-BASED PART MOBILITY ANALYSIS

Existing methods typically assume a known number of object parts to infer part mobility. In practice, they input the ground-truth part count to establish cross-state correspondences, which then serve as priors for clustering-based part segmentation. In contrast, our core idea is to understand the part mobility based on the motion cues in the interaction videos. Once the dual-Gaussian representation decouples the static base and dynamic components, we recover accurate, time-parameterised trajectories for the moving Gaussian primitives *over arbitrary time horizons with selectively sampled timesteps*. This enables motion-based part segmentation by clustering moving Gaussians with the same motion patterns into rigid parts. Therefore, as shown in Fig. 5, based on the clean inferred motion trajectories, we introduce a simple, robust, purely analysis-based sequential RANSAC (Magri & Fusiello, 2016) to achieve the part segmentation and estimate articulation parameters. Built on sequential RANSAC with Kabsch solver (Magri & Fusiello, 2016), AIM automatically recovers the part number and its kinematic parameters, *i.e.* joint type, axis direction, and motion magnitude.

**Part segmentation.** From the time-dependent moving Gaussian set $\{\mathcal{G}^M, t\}$ with learned deformation field $\mathcal{F}_\theta$, we can infer the centers positions of moving Guassian at timestep $t$ as $\mathcal{P}_t = \{\mu_{i,t}^M\}_{i=1}^{N_m}$. Furthermore, we can easily obtain the one-to-one corresponding trajectory between timestep $t$ and $t'$, recorded as $\{\mathcal{P}_{t \to t'}\}$. To extract rigid parts, we employ a sequential RANSAC with a Kabsch solver (Kabsch, 1976). Unlike conventional start-end matching methods ($t = 0 \ v.s. \ t = 1$) or pretrained segmentation-driven approaches, which require structural priors from manual input or pretrained models, our method aggregates evidence across trajectories spanning multiple time windows to capture diverse motions and improve robustness. For one trajectory of time window $\{\mathcal{P}_{a \to b}\}$, the optimal rigid transform is estimated by Kabsch solver, as

$$(\mathbf{R}_{a \to b}^*, \mathbf{t}_{a \to b}^*) = \arg\min_{\mathbf{R}, \mathbf{t}} \sum_{i \in \mathcal{S}_{\min}} \left\| \mu_{i,b}^M - (\mathbf{R}\mu_{i,a}^M + \mathbf{t}) \right\|^2, \tag{2}$$

where $\mathcal{S}_{\min}$ is a randomly sampled minimal set. The residual error of each moving Gaussian $\{\mathcal{G}_i^M\}$ is defined as:

$$\text{err}_i = \| \mu_{i,b}^M - (\mathbf{R}_{a \to b}^* \mu_{i,a}^M + \mathbf{t}_{a \to b}^*) \|. \tag{3}$$

A Gaussian $\mathcal{G}_i^M$ is accepted as an inlier if $\text{err}_i < \epsilon_{in}$. After $N_{\text{sampling}}$ iterations, the largest consensus set is selected as the support set. The motion parameters $(R, t)$ are subsequently re-estimated from all inliers using the Kabsch solver to obtain one motion hypothesis. The identified inliers are removed, and the process is repeated on the remaining Gaussians. The procedure terminates when no valid support set is found, when the maximum iteration budget $N_{\text{max\_iter}}$ is reached, or when the largest inlier set is small. This sequential RANSAC yields a collection of support sets, each corresponding to one rigid part. In this work, to balance accuracy and efficiency, we simultaneously employ the trajectories of two time windows, $\mathcal{P}_{0 \to 0.5}$ and $\mathcal{P}_{0 \to 1}$, and compute the mean residual error across them to determine inliers, as:

$$\text{err}_i = \frac{1}{2} \| \mu_{i,0.5}^M - (\mathbf{R}_{0 \to 0.5}^* \mu_{i,0}^M + \mathbf{t}_{0 \to 0.5}^*) \|$$
$$+ \frac{1}{2} \| \mu_{i,1}^M - (\mathbf{R}_{0 \to 1}^* \mu_{i,0}^M + \mathbf{t}_{0 \to 1}^*) \|. \tag{4}$$

**Per-part articulation parameters estimation.** With the selected support sets, we employ the Kabsch algorithm to estimate the rigid transformation $\{(\mathbf{R}_k, \mathbf{t}_k)\}_{k=1}^{K}$, $K$ is the number of support sets, *i.e.* the number of parts. Furthermore, we extract the underlying articulation parameters to characterise the motion. We follow existing works (Liu et al., 2023a; 2025; Weng et al., 2024) and focus on the following joint articulation parameters: the joint axis position $\mathbf{p}$, joint axis direction $\mathbf{u}$, translation distance $\Phi$, rotation angle $\Theta$, and joint type (prismatic or revolute). According to Rodrigues' rotation formula (Rodrigues, 1840), the rotation matrix $\mathbf{R}_k$ can be expressed as:

$$\mathbf{R}_k = \cos\Theta_k \mathbf{I} + \sin\Theta_k [\mathbf{u}_k]_\times + (1 - \cos\Theta_k)(\mathbf{u}_k \otimes \mathbf{u}_k), \tag{5}$$

where direction $\mathbf{u}_k$ is a unit vector, $\mathbf{I}$ is the identity matrix, $[\cdot]_\times$ is the cross product and $\otimes$ is the outer product. From Eq. 5, we can obtain the $\mathbf{u}_k$ and $\Theta_k$, respectively, as:

$$\mathbf{u}_k = \frac{1}{2\sin\Theta_k} \begin{pmatrix} \mathbf{R}_k[2,1] - \mathbf{R}_k[1,2] \\ \mathbf{R}_k[0,2] - \mathbf{R}_k[2,0] \\ \mathbf{R}_k[1,0] - \mathbf{R}_k[0,1] \end{pmatrix}, \quad \Theta_k = \arccos\left(\frac{\mathrm{tr}(\mathbf{R}_k) - 1}{2}\right). \tag{6}$$

For the translation distance $\Phi$ and the position $\mathbf{p}$ (start point) of joint axis, we can calculate them based on the rotation matrix $\mathbf{R}_k$ and translation component $\mathbf{t}_k$ as:

$$\Phi_k = \left|\frac{\mathbf{u}_k \cdot \mathbf{t}_k}{\|\mathbf{u}_k\|^2}\right|, \quad \mathbf{p}_k = (\mathbf{R}_k - \mathbf{I})^{-1} \cdot (\Phi_k \cdot \mathbf{u}_k - \mathbf{t}_k). \tag{7}$$

For the joint type, inspired by (Shi et al., 2021; Liu et al., 2025), we classify it as revolute when the rotation degree $\Theta$ exceeds a threshold $\epsilon_{revol} = 10°$ (about 0.17 radians) and prismatic by contrast. Based on this, in static-during-motion detection, a region in a moving Gaussian set is considered static if the rotation angle $\Theta \leq 0.1$ radians and translation magnitude $\Phi \leq 0.05$ units.

## 4 EXPERIMENT

### 4.1 DATASET, METRICS, AND IMPLEMENTATION

**Dataset.** We select various articulated objects from *PartNet-Mobility (Mo et al., 2019)*. We rendered a video of articulated motion using a camera trajectory around the object. To verify the effectiveness of our prior-free part segmentation, we render objects with multiple parts moving simultaneously in a variety of motions. For two-state baselines, we render 100 random upper-hemisphere views of the start and end states, respectively. Our benchmarks include challenging 8 two-part objects, 2 three-part objects, and 2 multi-part objects. Most interior parts are gradually revealed over time, reflecting real-world applications. (***Additional examples are provided in the Appendix C.***)

**Metrics.** We conduct comparisons from three perspectives: 1) **Part segmentation performance.** To consider the points inside the surface, we voxelize the meshes and evaluate part-level segmentation with *3D Intersection-over-Union (IoU)* (Nie et al., 2021); 2) **Reconstruction quality.** Following prior works, we compute the *bi-directional Chamfer Distance (mm)* to measure the reconstruction quality; 3) **Articulation estimation accuracy**. Following prior works, We report the *Axis Ang Err (°)*, *Axis Pos Err (0.1m)*, and *Part Motion (° or m)*. (***More details please refer to Appendix B***).

**Implementation details.** We evaluate against recent state-of-the-art methods, PARIS [†], DTA, and ArtGS, that use RGB–Depth inputs. Our approach requires an RGB video with 200 frames for two- and three-part objects and 500 frames for more complex objects. We present a baseline, *Ours-b*, which replaces the proposed dual-Gaussian representation with a single deformable 3D Gaussian shape (3DGS). Following ArtGS, we use a truncated signed distance function (TSDF) volume for mesh reconstruction, render depth maps, and marching cubes (Huang et al., 2024a).

### 4.2 QUALITATIVE AND QUANTITATIVE EVALUATION

**Part segmentation.** As shown in Tab. 1, our method attains the best 3D IoU on almost all objects in both two-part and multi-part settings. On complex objects, the gains are large, *e.g.* on *Storage* (6 moving parts), our mean dynamic-part IoU exceeds the prior SoTA by ***+27.11***%. Standard deviations are consistently lower, indicating greater stability than two-state inference (*e.g.* DTA/ArtGS on *Fridge*, *Oven*). Compared with *Ours-b*, the dual-Gaussian dynamic–static separation further improves accuracy by suppressing static interference.

---

[†]PARIS is augmented with depth supervision

Table 1: **Part segmentation performance on articulated objects.** (a) Two-part; (b) Three-part; (c) Complex objects. For two-part objects, 3D IoU(%) is reported as mean±std over 10 trials, while for three-part and complex objects, we report mean 3D IoU(%) over 10 trials. $D_{avg}$ represents the average over all movable parts. $F$ denotes failure. Higher is better, with the best highlighted in **bold**.; Gray means two-state methods.

(a) Two-part objects

| 3D IoU (%)↑ | Method | Two-part objects | | | | | | |
|---|---|---|---|---|---|---|---|---|
| | | Fridge | Oven | Scissor | USB | Washer | Blade | Storage |
| Static Part | PARIS | $85.23_{\pm9.20}$ | $92.19_{\pm6.33}$ | $83.13_{\pm3.71}$ | F | $\mathbf{98.53_{\pm0.48}}$ | $\mathbf{87.84_{\pm2.60}}$ | $86.73_{\pm3.18}$ |
| | DTA | $86.27_{\pm6.58}$ | $91.11_{\pm3.50}$ | $66.01_{\pm3.38}$ | $86.30_{\pm1.84}$ | $88.49_{\pm10.94}$ | $83.22_{\pm2.03}$ | $88.87_{\pm1.84}$ |
| | ArtGS | $87.69_{\pm10.46}$ | $94.55_{\pm3.76}$ | $84.70_{\pm4.27}$ | $88.62_{\pm2.32}$ | $94.51_{\pm4.04}$ | $90.91_{\pm1.57}$ | $88.68_{\pm2.66}$ |
| | Ours-b | $83.70_{\pm4.70}$ | $94.46_{\pm2.31}$ | $81.27_{\pm2.84}$ | $55.15_{\pm14.96}$ | F | $83.59_{\pm1.41}$ | $88.57_{\pm3.07}$ |
| | Ours | $\mathbf{88.01_{\pm6.37}}$ | $\mathbf{97.72_{\pm0.52}}$ | $\mathbf{92.42_{\pm1.05}}$ | $\mathbf{92.93_{\pm1.30}}$ | $96.98_{\pm2.22}$ | $84.33_{\pm1.65}$ | $\mathbf{91.41_{\pm2.78}}$ |
| Dynamic Part | PARIS | $55.97_{\pm29.62}$ | $45.42_{\pm43.90}$ | $67.81_{\pm15.09}$ | F | $32.75_{\pm29.62}$ | $34.42_{\pm10.85}$ | $42.88_{\pm16.99}$ |
| | DTA | $52.06_{\pm22.22}$ | $41.23_{\pm22.06}$ | $53.58_{\pm7.44}$ | $79.81_{\pm2.71}$ | $5.97_{\pm5.64}$ | $27.92_{\pm11.62}$ | $39.01_{\pm10.74}$ |
| | ArtGS | $58.78_{\pm36.40}$ | $65.68_{\pm24.50}$ | $75.04_{\pm9.69}$ | $86.85_{\pm2.37}$ | $35.62_{\pm36.43}$ | $28.95_{\pm16.08}$ | $65.30_{\pm9.66}$ |
| | Ours-b | $57.09_{\pm17.60}$ | $72.50_{\pm9.45}$ | $77.74_{\pm3.75}$ | $55.36_{\pm14.52}$ | F | $41.76_{\pm3.34}$ | $35.54_{\pm26.04}$ |
| | Ours | $\mathbf{75.19_{\pm14.61}}$ | $\mathbf{89.61_{\pm1.50}}$ | $\mathbf{92.21_{\pm1.19}}$ | $\mathbf{91.95_{\pm1.18}}$ | $\mathbf{68.52_{\pm13.80}}$ | $\mathbf{43.92_{\pm6.97}}$ | $\mathbf{69.01_{\pm7.42}}$ |
| Mean | PARIS | $70.60_{\pm19.40}$ | $68.80_{\pm25.11}$ | $75.47_{\pm9.01}$ | F | $65.64_{\pm14.34}$ | $61.13_{\pm6.64}$ | $64.81_{\pm10.08}$ |
| | DTA | $69.16_{\pm13.56}$ | $66.17_{\pm12.75}$ | $59.79_{\pm5.41}$ | $83.05_{\pm2.26}$ | $47.23_{\pm6.89}$ | $55.57_{\pm6.80}$ | $63.94_{\pm4.57}$ |
| | ArtGS | $73.24_{\pm23.42}$ | $80.12_{\pm14.13}$ | $79.87_{\pm6.19}$ | $87.73_{\pm2.10}$ | $65.06_{\pm20.21}$ | $59.93_{\pm8.82}$ | $76.99_{\pm6.17}$ |
| | Ours-b | $70.40_{\pm11.13}$ | $83.48_{\pm5.86}$ | $79.51_{\pm3.26}$ | $55.26_{\pm14.51}$ | F | $62.68_{\pm2.47}$ | $62.06_{\pm13.79}$ |
| | Ours | $\mathbf{81.60_{\pm10.48}}$ | $\mathbf{93.66_{\pm0.98}}$ | $\mathbf{92.31_{\pm1.12}}$ | $\mathbf{92.44_{\pm1.24}}$ | $\mathbf{82.75_{\pm7.99}}$ | $\mathbf{64.12_{\pm4.29}}$ | $\mathbf{80.21_{\pm7.37}}$ |

(b) Three-part objects

| | 3D IoU | PARIS-m | DTA | ArtGS | Ours |
|---|---|---|---|---|---|
| Storage 47024 | $S$ | 89.77 | 88.53 | 92.23 | **94.20** |
| | $D_0$ | F | 55.32 | 51.78 | **94.95** |
| | $D_1$ | | 60.72 | **96.00** | 79.75 |
| Fridge 11304 | $S$ | 89.55 | 81.05 | 94.80 | **95.42** |
| | $D_0$ | 25.73 | 55.28 | 88.53 | **89.95** |
| | $D_1$ | 45.31 | 61.19 | 80.48 | **91.42** |

(c) Complex objects

| | 3D IoU (%) | DTA | ArtGS | Ours |
|---|---|---|---|---|
| Storage 47648 | $S$ | 87.95 | 93.32 | **97.01** |
| | $D_{avg}$ | 26.38 | 52.23 | **79.34** |
| | Mean | 35.18 | 58.14 | **81.87** |
| Table 31249 | $S$ | 89.81 | 90.44 | **91.75** |
| | $D_{avg}$ | 37.29 | 38.07 | **43.92** |
| | Mean | 47.80 | 48.55 | **53.49** |

Table 2: **Mesh reconstruction comparison.** (a) Two-part objects; (b) Three-part objects; (c) Complex objects. For two-part objects, we report CD distance (mm) as mean$\pm_{std}$ across 10 trials. For three-part and complex objects, we only report the mean value, while we report average CD for movable parts. Lower (↓) is better.

(a) Two-part objects

| Metric | Method | Two-part objects | | | | | | |
|---|---|---|---|---|---|---|---|---|
| | | Fridge | Oven | Scissor | USB | Washer | Blade | Storage |
| CD-S | DTA | $3.19_{\pm0.80}$ | $9.10_{\pm3.59}$ | $9.41_{\pm1.00}$ | $2.04_{\pm0.12}$ | $\mathbf{5.03_{\pm3.44}}$ | $\mathbf{0.33_{\pm0.00}}$ | $4.94_{\pm0.14}$ |
| | ArtGS | $\mathbf{1.58_{\pm0.28}}$ | $\mathbf{8.39_{\pm0.29}}$ | $0.80_{\pm0.99}$ | $11.01_{\pm0.43}$ | $6.63_{\pm0.17}$ | $1.23_{\pm0.01}$ | $7.50_{\pm0.15}$ |
| | Ours-b | $4.73_{\pm0.53}$ | $10.08_{\pm1.43}$ | $2.22_{\pm1.07}$ | $34.61_{\pm1.73}$ | F | $1.90_{\pm0.06}$ | $7.27_{\pm0.78}$ |
| | Ours | $3.45_{\pm0.09}$ | $10.36_{\pm0.73}$ | $\mathbf{0.14_{\pm0.00}}$ | $\mathbf{1.54_{\pm0.14}}$ | $9.25_{\pm0.99}$ | $1.76_{\pm0.02}$ | $\mathbf{7.09_{\pm0.49}}$ |
| CD-m | DTA | $4.08_{\pm0.60}$ | $77.61_{\pm86.59}$ | $141.99_{\pm35.39}$ | $1.90_{\pm0.51}$ | $481.06_{\pm66.34}$ | $19.30_{\pm2.09}$ | $67.33_{\pm3.03}$ |
| | ArtGS | $43.51_{\pm0.47}$ | $64.34_{\pm3.66}$ | $53.71_{\pm28.82}$ | $50.00_{\pm19.77}$ | $155.65_{\pm22.79}$ | $473.72_{\pm7.62}$ | $6.92_{\pm0.54}$ |
| | Ours-b | $18.27_{\pm4.22}$ | $5.17_{\pm2.54}$ | $73.13_{\pm50.33}$ | $19.46_{\pm0.46}$ | F | $110.46_{\pm36.00}$ | $88.85_{\pm73.10}$ |
| | Ours | $\mathbf{2.21_{\pm0.18}}$ | $\mathbf{1.63_{\pm0.25}}$ | $\mathbf{0.27_{\pm0.03}}$ | $\mathbf{0.89_{\pm0.10}}$ | $\mathbf{21.03_{\pm1.02}}$ | $\mathbf{2.36_{\pm0.09}}$ | $\mathbf{18.95_{\pm2.57}}$ |

(b) Three-part objects

| | ↓ | PARIS-m | DTA | ArtGS | Ours |
|---|---|---|---|---|---|
| Storage 47024 | CD-s | 8.05 | **3.20** | 3.58 | 10.37 |
| | CD-$D_0$ | F | 275.87 | 253.69 | **0.81** |
| | CD-$D_1$ | | 287.70 | **1.21** | 27.35 |
| Fridge 11304 | CD-s | 6.91 | 4.90 | **2.93** | 8.16 |
| | CD-$D_0$ | 298.29 | 29.95 | 12.83 | **3.85** |
| | CD-$D_1$ | 189.85 | 323.06 | 12.17 | **2.12** |

(c) Complex objects

| | ↓ | DTA | ArtGS | Ours |
|---|---|---|---|---|
| Storage 47648 | CD-s | **2.08** | 2.96 | 2.63 |
| | CD-m$_{avg}$ | 200.15 | 71.17 | **8.36** |
| Table 31249 | CD-s | **2.56** | 3.65 | 3.08 |
| | CD-m$_{avg}$ | 152.93 | 51.40 | **4.99** |

**Mesh reconstruction.** We report mesh-reconstruction results in Tab. 2. Despite using ***RGB-only*** inputs, our CD on static parts is competitive with PARIS and ArtGS, and our errors on dynamic parts are much lower *e.g. Storage*$_{47648}$: 8.36 vs. 71.17 (ArtGS); *Table*: 4.99 vs. 51.40.

**Articulation estimation.** Our framework attains highly accurate joint predictions (see Tab. 3). For two-part objects, our axis-angle errors are consistently minimal (*e.g. Oven*: 0.27° vs. 5.39° of DTA). For complex objects, the improvements are striking: on *Storage*, we reduce axis-angle error from 12.78° (ArtGS) to 0.58°, and part motion errors to nearly zero (0.02 for prismatic joints). This evidences the advantage of dual-Gaussian representation and motion-based fitting.

**Analysis of two-state limitations.** From qualitative and quantitative results, it can be observed that two-state methods strongly rely on geometric correspondence between the start and end states. Once this correspondence is broken, such as the open-end state reveals interior regions absent from the close-start, these methods are forced to match dissimilar geometry, leading to degraded part segmentation and unstable articulation estimation. Most notably, on two-part objects such as the fridge and oven in Fig. 6, the newly revealed interior structures cause the canonical mid-state Gaussian initialisation in ArtGS to fail. This disturbed canonical initialisation propagates to articulation estimation

Table 3: **Quantitative evaluation of articulation estimation.** (a) Two-part; (b) Three-part; (c) Complex objects. For complex objects, we report the average of all moving parts. Due to the different magnitudes of part motion for revolute and prismatic joints, we report both of them. $F$ denotes failure. $WT$ denotes that more than 6 out of 10 trials result in an incorrect joint-type prediction. $-$ indicates prismatic joints w/o rotation axis.

(a) Two-part objects

| Metric | Method | Two-part objects | | | | | | |
|---|---|---|---|---|---|---|---|---|
| | | Fridge | Oven | Scissor | USB | Washer | Blade | Storage |
| Axis Ang | DTA | $1.86_{\pm 3.80}$ | $5.39_{\pm 7.16}$ | $1.01_{\pm 1.23}$ | $0.22_{\pm 0.11}$ | $17.34_{\pm 25.59}$ | $1.65_{\pm 0.35}$ | $8.18_{\pm 6.80}$ |
| | ArtGS | WT | WT | $2.39_{\pm 1.91}$ | $23.86_{\pm 35.02}$ | WT | $1.31_{\pm 0.14}$ | $0.00_{\pm 0.01}$ |
| | Ours-b | $6.76_{\pm 3.40}$ | $3.36_{\pm 3.31}$ | $5.12_{\pm 0.46}$ | $6.65_{\pm 4.20}$ | F | $0.25_{\pm 4.20}$ | $1.72_{\pm 0.67}$ |
| | Ours | $2.70_{\pm 1.73}$ | $0.27_{\pm 0.25}$ | $1.60_{\pm 0.38}$ | $0.59_{\pm 0.30}$ | $1.63_{\pm 0.90}$ | $0.18_{\pm 0.17}$ | $1.52_{\pm 0.88}$ |
| Axis Pos | DTA | $1.75_{\pm 1.20}$ | $4.98_{\pm 4.38}$ | $8.84_{\pm 4.76}$ | $0.01_{\pm 0.01}$ | $26.50_{\pm 42.41}$ | $-$ | $-$ |
| | ArtGS | WT | WT | $1.73_{\pm 1.70}$ | $6.01_{\pm 8.60}$ | WT | $-$ | $-$ |
| | Ours-b | $1.22_{\pm 0.86}$ | $1.24_{\pm 0.86}$ | $0.86_{\pm 0.55}$ | $0.84_{\pm 0.37}$ | F | $-$ | $-$ |
| | Ours | $0.86_{\pm 0.34}$ | $1.13_{\pm 0.68}$ | $0.75_{\pm 0.05}$ | $1.45_{\pm 0.71}$ | $1.12_{\pm 0.29}$ | $-$ | $-$ |
| Part Motion | PARIS | $167.60_{\pm 14.49}$ | $144.80_{\pm 11.20}$ | $122.05_{\pm 42.50}$ | F | $86.13_{\pm 3.11}$ | $0.08_{\pm 0.06}$ | $0.10_{\pm 0.02}$ |
| | DTA | $171.77_{\pm 13.43}$ | $142.10_{\pm 33.62}$ | $150.50_{\pm 11.29}$ | $0.32_{\pm 0.15}$ | $76.43_{\pm 11.27}$ | $0.02_{\pm 0.06}$ | $0.07_{\pm 0.06}$ |
| | ArtGS | WT | WT | $99.09_{\pm 67.18}$ | $120.05_{\pm 19.63}$ | WT | $0.14_{\pm 0.00}$ | $0.00_{\pm 0.00}$ |
| | Ours-b | $10.67_{\pm 3.49}$ | $7.64_{\pm 1.77}$ | $5.20_{\pm 0.40}$ | $16.94_{\pm 1.85}$ | F | $0.25_{\pm 4.20}$ | $1.72_{\pm 0.67}$ |
| | Ours | $6.76_{\pm 3.40}$ | $3.36_{\pm 3.31}$ | $5.12_{\pm 0.46}$ | $6.65_{\pm 4.20}$ | $6.90_{\pm 2.88}$ | $0.01_{\pm 0.00}$ | $0.02_{\pm 0.00}$ |

(b) Three-part objects

| | Methods | $D_0$ | | | $D_1$ | | |
|---|---|---|---|---|---|---|---|
| | | Axis Ang | Axis Pos | Part Motion | Axis Ang | Axis Pos | Part Motion |
| Storage 47024 | DTA | 58.63 | 38.59 | 96.56 | **0.50** | $-$ | 0.01 |
| | ArtGS | 20.63 | 3.83 | 107.56 | 1.75 | $-$ | 0.13 |
| | Ours | **0.56** | **1.26** | **1.66** | 1.03 | $-$ | 0.06 |
| Fridge 11304 | DTA | 22.02 | 359.06 | 178.80 | 9.48 | 6.22 | 38.36 |
| | ArtGS | 13.94 | 46.95 | 176.52 | 3.33 | 15.79 | 41.81 |
| | Ours | **0.68** | **3.58** | **3.57** | **1.67** | **0.68** | **4.81** |

(c) Complex objects

| | | ↓ | $Ang_{avg}$ | $Pose_{avg}$ | $Motion^r_{avg}$ | $Motion^p_{avg}$ |
|---|---|---|---|---|---|---|
| Storage 47648 | ArtGS | | 12.78 | 3.34 | 81.93 | 0.18 |
| | Ours | | **0.58** | **1.31** | **10.56** | **0.02** |
| Table 31249 | ArtGS | | 33.19 | 2.42 | 82.29 | 0.43 |
| | Ours | | **1.19** | **0.81** | **1.10** | **0.01** |

Figure 6: Qualitative results of part segmentation and articulation estimation on two two-part objects (fridge, left; oven, middle) and a complex multi-part object (Storage-47648, right). For complex object, each predicted joint axis is visualised using the same colour as its corresponding part segmentation mask. Across the two-part objects, DTA and ArtGS often struggle with mis-segmentation and inaccurate joint-axis/type predictions. In contrast, our method produces clean part segmentation and consistent joint-axis estimation across all objects.

and results in joint-type errors, often predicting a prismatic joint instead of the correct revolute joint. By contrast, DTA is relatively more stable due to its symmetric optimisation of both the start-to-end and end-to-start transformations, yet it still misclassifies newly revealed structures, *e.g.* predicting the oven interior as dynamic or assigning large portions of the moving fridge door to the static part.

**Summary.** Overall, compared to two-state motion inference, our method demonstrates stronger and more stable performance under a challenging close-start/open-end setting. In the two- and three-part datasets, *Articulation in Motion* achieves the best results on the vast majority of objects. On complex objects, our approach shows a clear advantage, significantly surpassing prior methods *(more visualisations in Appendix D and Fig. A4)*. For mesh reconstruction, while our approach still has limitations in overall mesh fidelity compared with NeRF-based methods, the strength of our part mobility analysis enables consistently superior reconstruction of dynamic parts over prior state-of-the-art methods.

Table 4: Ablation studies on complex objects. We report the average metrics of dynamic parts. And we calculate the mean across three trials.

| | | $Ang_{avg}\downarrow$ | $Pose_{avg}\downarrow$ | $Motion^r_{avg}\downarrow$ | $Motion^p_{avg}\downarrow$ | $CD\text{-}m_{avg}\downarrow$ | $3D\ IoU^D_{avg}\uparrow$ |
|---|---|---|---|---|---|---|---|
| | ArtGS | 12.78 | 3.34 | 81.93 | 0.18 | 71.17 | 52.23 |
| | $w/o$ start state scan | 1.57 | 1.49 | 20.61 | 0.05 | 97.65 | 37.60 |
| Storage | $w/o$ SDMD | 2.62 | 1.41 | 14.72 | 0.06 | 91.52 | 77.45 |
| *47648* | $w/o$ dual-GS | 2.95 | 1.77 | 15.36 | 0.08 | 17.43 | 67.66 |
| | $w/o$ RANSAC | 3.80 | 1.41 | 12.62 | 0.38 | 78.54 | 67.06 |
| | Full | 0.58 | 1.31 | 10.56 | 0.02 | 8.36 | 79.34 |
| | ArtGS | 33.19 | 2.42 | 82.29 | 0.43 | 51.40 | 38.07 |
| | $w/o$ start state scan | | | F | | | |
| Table | $w/o$ SDMD | 11.62 | 1.49 | 23.74 | 0.21 | 18.05 | 37.10 |
| *31249* | $w/o$ dual-GS | 1.49 | 0.94 | 1.47 | 0.37 | 6.26 | 41.74 |
| | $w/o$ RANSAC | 1.28 | 0.91 | 1.25 | 0.37 | 6.02 | 40.65 |
| | Full | 1.19 | 0.81 | 1.10 | 0.01 | 4.99 | 43.92 |

## 4.3 ABLATION STUDY

**Effectiveness of start state scan.** As shown in Tab. 4, while directly training the dual-Gaussian representation with a set of random Gaussians as the static, 3D IoU$^D avg$ drops from **79.34**% to **37.60**%. On *Table*, the pipeline cannot detect the moving Gaussians. These results indicate that the start state can anchor the shape and appearance of objects and is essential for capturing motion cues.

**Effectiveness of the static-during-motion detection (SDMD).** Disabling SDMD consistently harms dynamic geometry and motion recovery. In particular, the sharp increase in CD-m$^D_{avg}$ in both storage and table shows that the filtering of static noise during motion capture is critical to part mobility analysis. *(More visual results please see Fig. A13 in Appendix.)*

**Effectiveness of the dual-GS representation.** We assess the dual-Gaussian representation by replacing it with the original deformable-3DGS. Across metrics, articulation accuracy and part-segmentation IoU degrade markedly without our dual-Gaussian representation. This confirms that explicit dynamic–static disentanglement is a cornerstone for prior-free part mobility analysis. *(More visual results please see Fig. A14 in Appendix.)*

**Effectiveness of sequential RANSAC.** We first attempted prior-free density clustering with DB-SCAN (Ester et al., 1996), which failed to produce valid partitions across objects. We then applied K-means (Pham et al., 2004) with the provided part count, yielding reasonable groups but inferior articulation and segmentation. In contrast, sequential RANSAC delivers the best prior-free performance while remaining robust to motion variability. Given our accurate motion trajectories, K-means outperforms ArtGS, underscoring the quality of our motion cue extraction.

## 5 CONCLUSION AND LIMITATION

**Conclusion.** In this work, we presented a compact pipeline, *Articulation in Motion* (AIM), to achieve a prior-free and stable part-mobility analysis. Compared to previous works based on two-state shape correspondence, our method utilises more natural motion and human-object interaction videos as input. It introduced a dual-Gaussian scene representation to analyse motion cues in the video. With the dual-Gaussian dynamic–static separation, we obtained clean motion trajectories; coupled with the robustness of sequential RANSAC, this enables prior-free part segmentation and articulation on unseen objects. Comprehensive experimental evaluations validated the effectiveness and stability of the proposed AIM in diverse challenging scenarios.

**Limitations and future work.** Our AIM generates higher quality segmentation of articulated objects and recovery of their degrees of freedom (DoF) compared to prior work. We do not require the use of depth sensing, as done by most prior art techniques; however, we do utilise a video that captures the DoFs of the object's parts. Such a video contains more information than a three-dimensional reconstruction of the object at the start and end states. Yet, in many common cases, this is easier to capture compared to the former type of data: some objects contain many DoFs, and some are dependent on each other, making it hard to capture all of them with only two static states. The capture of a video is a more natural way for a person to introduce an object, where they can expose each DoF at a time. Capturing the whole geometry of internal parts, such as drawers, an extended blade of a knife, or the blades of a pair of scissors, requires the full disassembly of the articulated object. The generated geometry is limited to the visible geometry and, as such, may be limited in its application for developing interactive models. Future work may utilise a data-driven approach to complete such parts of the whole geometry, given a dataset of these parts.

ACKNOWLEDGMENTS

This project is supported by the Amazon Research Award "PCo3D: Physically Plausible Controllable 3D Generative Models". The project is also supported by an unrestricted charitable donation from Meta (Aria Project Partnership) to the University of Birmingham. The computations in this research were supported by the Baskerville Tier 2 HPC service. Baskerville was funded by the EPSRC and UKRI through the World Class Labs scheme (EP\T022221\1) and the Digital Research Infrastructure programme (EP\W032244\1) and is operated by Advanced Research Computing at the University of Birmingham.

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

APPENDIX

This is the appendix for the main paper. Here is a general roadmap describing the contents of each part of this document supporting the main paper:

- In Section A, we first review the core 3DGS (Kerbl et al., 2023) and Deformable 3DGS (Yang et al., 2024) formulations to unify notation and provide the foundational baseline for our dual-Gaussian representation in the main paper.

- In Section B, we detail three evaluation aspects: part segmentation, reconstruction quality, and articulation estimation accuracy .

- In Section C, we provide additional details to our rendered datasets, which include the detailed splits/statistics and representative multi-view examples.

- Section D provides more experimental results, including more visual comparisons among DTA (Weng et al., 2024), ArtGS (Liu et al., 2025) and ours (Section D.1), more quantitative and qualitative results compared with pre-trained segmentation driven method Video2Articulation (Peng et al., 2025) (Section D.2), more results compared with DTA and ArtGS under the open-start and open-end setting (Section D.3). Sec. D.4 details our real-world data acquisition and preprocessing pipeline, and reports visual results.

- Section E provides additional qualitative results for the ablation study.

## A  PRELIMINARY

**3D Gausssian splatting** 3DGS represents the scene as an explicit point-based 3D structure, enabling orders of magnitude faster reconstruction and rendering. In this work, we build on 3DGS to reconstruct the articulated objects with the part-level structures. In details, each 3D Gaussian $\mathcal{G}_i$ is defined by a center position $\mu_i \in \mathbb{R}^3$, opacity $\sigma_i \in \mathbb{R}$, a covariance matrix $\Sigma_i$ parameterized by a 3D scale $s_i \in \mathbb{R}^3$ and a rotation $r_i \in \mathbb{R}^4$, as well as spherical harmonics (SH) coefficients $h_i$ for view-dependent color modeling (Kerbl et al., 2023). Given image captures of the scene, we optimise a collection of Gaussians $\{\mathcal{G}\} = \{\mathcal{G}_i\}_{i=1}^N$ via blending-based differentiable rendering:

$$\mathcal{I} = \sum_{i=1}^N \mathbf{c}_i T_i \alpha_i^{2D}, \ \alpha_i^{2D}(u) = \sigma_i \exp(-\frac{1}{2}(u - \mu_i^{2D})^T \Sigma_i^{2D^{-1}}(u - \mu_i^{2D})), \ T_i = \prod_{j=1}^{i-1}(1 - \alpha_j^{2D}), \ (8)$$

where $\mu_i^{2D}$ and $\Sigma_i^{2D}$ denote the 2D projections of the 3D center $\mu_i$ and covariance matrix $\Sigma_i$, respectively; $u$ represents the pixel coordinate; and $\mathbf{c}_i$ is the colour of $\mathcal{G}_i$, determined by the SH coefficients $h_i$ and the view direction. $T_i$ represents the transmittance from the start of rendering to $\mathcal{G}_i$.

**Deformable 3D Gaussian splatting.** To model the time-varying changes of geometry and appearance in a dynamic scene, Deformable-3DGS (Yang et al., 2024) introduces a learnable deformation field to model temporal transformations in the centre positions $\mu$, rotations $r$, and scales $s$ of 3D Gaussians. This deformation field is parameterised by a multi-layer perceptron (MLP), which predicts offsets based on time and the canonical Gaussian $\mathcal{G}_c$. Specifically, at time step $t$, the deformed Gaussian $\mathcal{G}_d$ is defined as:

$$\mathcal{G}_d(\mu_i, s_i, r_i, \sigma_i, h_i) = \mathcal{G}_c(\mu_i + \delta\mu, s_i + \delta s, r_i + \delta r, \sigma_i, h_i), \quad (9)$$

where the offsets $\delta\mu, \delta s, \delta r$ are given by $F_\theta(\gamma(t), \gamma(\mu_i))$, with $F_\theta$ representing the deformation field and $\gamma$ denoting the positional encoding function. With differentiable rendering, both the Gaussian parameters and the deformation network parameters are jointly optimised. Here, we only consider the time-varying transformation of position $\delta\mu$ and rotation $\delta r$.

## B  METRICS AND EVALUATION DETAILS

We evaluate from three perspectives consistent with the main text: (1) *part segmentation* via 3D IoU on voxelized meshes, (2) *reconstruction quality* via bi-directional Chamfer Distance (mm), and (3) *articulation estimation accuracy* via axis and motion errors.

### 1. PART SEGMENTATION PERFORMANCE (3D IoU)

For each predicted part and its ground-truth (GT) counterpart, we voxelize both meshes onto a shared binary occupancy grid (identical bounds and voxel size). Let $\mathcal{V}^p$ and $\mathcal{V}^g$ be the sets of occupied voxels. The part-level segmentation score is

$$\text{IoU} = \frac{|\mathcal{V}^p \cap \mathcal{V}^g|}{|\mathcal{V}^p \cup \mathcal{V}^g|}, \tag{10}$$

as in prior work (Nie et al., 2021).

### 2. RECONSTRUCTION QUALITY (CHAMFER DISTANCE, MM)

We uniformly sample points on the surfaces of the predicted and GT meshes and compute the symmetric (bi-directional) Chamfer Distance in millimetres. For point sets $X$ and $Y$,

$$\text{CD}(X, Y) = \frac{1}{|X|} \sum_{x \in X} \min_{y \in Y} \|x - y\|_2 + \frac{1}{|Y|} \sum_{y \in Y} \min_{x \in X} \|y - x\|_2. \tag{11}$$

We report CD for the whole object (*CD-w*), the static parts (*CD-s*), and the movable parts (*CD-m*).

### 3. ARTICULATION ESTIMATION ACCURACY

For each dynamic joint, we report three metrics:

**Axis Ang Err (°).** Let $\hat{\boldsymbol{a}}^p, \hat{\boldsymbol{a}}^g \in \mathbb{R}^3$ be the unit axis directions (predicted and GT). The angular error (orientation-invariant) is

$$\theta = \min\left\{ \arccos(\hat{\boldsymbol{a}}^p \cdot \hat{\boldsymbol{a}}^g),\ 180° - \arccos(\hat{\boldsymbol{a}}^p \cdot \hat{\boldsymbol{a}}^g) \right\}. \tag{12}$$

**Axis Pos Err (0.1m).** Let the axes be lines $\ell^p(s) = \boldsymbol{o}^p + s\,\hat{\boldsymbol{a}}^p$ and $\ell^g(t) = \boldsymbol{o}^g + t\,\hat{\boldsymbol{a}}^g$. The shortest distance $d$ between two 3D lines is used, and we report it in units of $0.1m$ by

$$\text{AxisPosErr} = 10 \times d, \qquad d = \frac{\left|(\hat{\boldsymbol{a}}^p \times \hat{\boldsymbol{a}}^g) \cdot (\boldsymbol{o}^p - \boldsymbol{o}^g)\right|}{\|\hat{\boldsymbol{a}}^p \times \hat{\boldsymbol{a}}^g\|} \quad \text{(for non-parallel axes)}, \tag{13}$$

and $d = \|(\boldsymbol{o}^g - \boldsymbol{o}^p) \times \hat{\boldsymbol{a}}^p\|$ for (nearly) parallel axes. This metric is reported for *revolute* joints only.

**Part Motion (° or m).** Between the start state $t=0$ and end state $t=1$, we measure the state error: (i) *revolute*: geodesic angle on SO(3) between the predicted and GT relative rotations $\Delta R^p = R_1^p (R_0^p)^\top$ and $\Delta R^g = R_1^g (R_0^g)^\top$,

$$\phi = \arccos\left( \frac{\text{tr}(\Delta R^p (\Delta R^g)^\top) - 1}{2} \right) \cdot \frac{180°}{\pi}; \tag{14}$$

(ii) *prismatic*: Euclidean difference between relative translations $\Delta t^p = t_1^p - t_0^p$ and $\Delta t^g = t_1^g - t_0^g$,

$$s = \|\Delta t^p - \Delta t^g\|_2 \quad \text{(meters)}. \tag{15}$$

## C  DATASETS

### C.1  SYNTHETIC DATASET

We build a dataset to evaluate motion segmentation under increasing difficulty. Detailed splits and statistics are reported in Table A1, Table A2, and Table A3. Each scene contains a start (static) state, a continuous motion segment, and an end (static) state. We also provide visual examples from the dataset. Fig. A1 shows a two-object scene with 100 multi-view images for the start state, 200 for the motion segment. Fig. A2 shows a three-object scene with the same counts: 100/200. Fig. A3 is a complex object scene with a longer motion segment: 100/500.

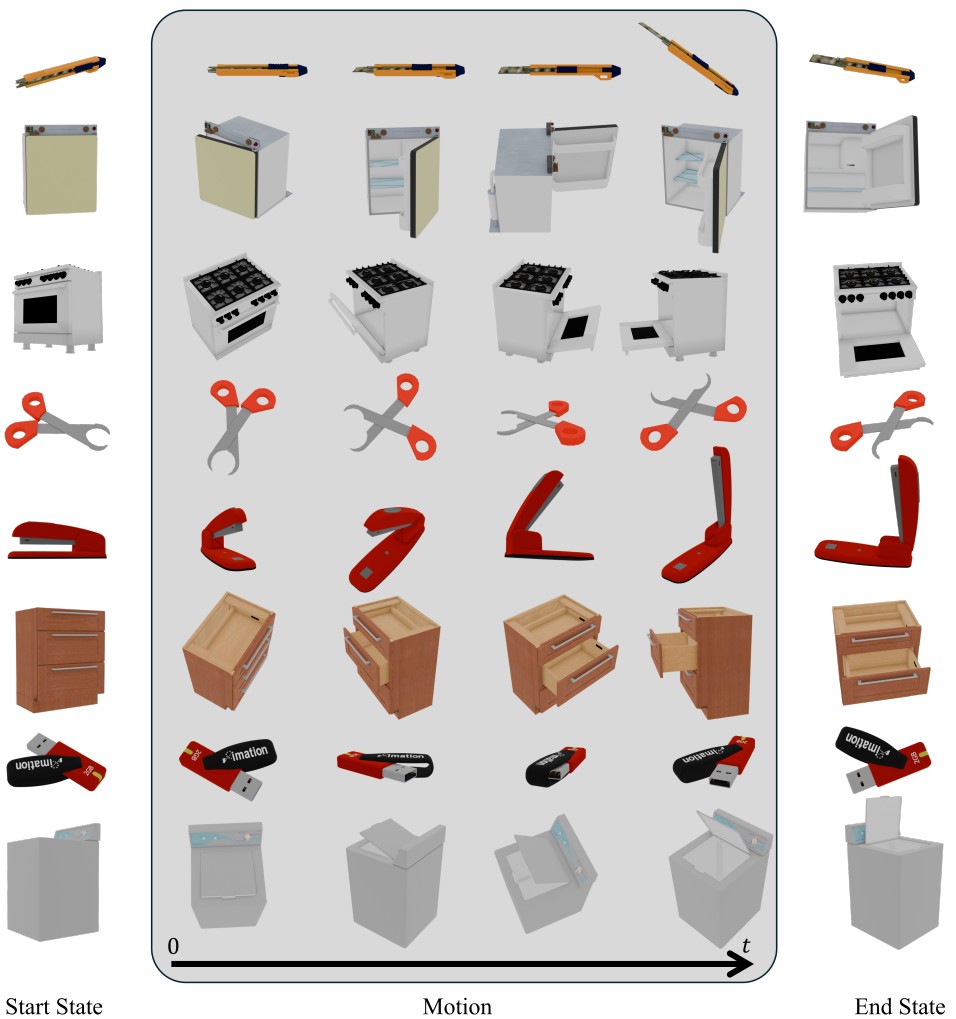

Start State    Motion    End State

Figure A1: Examples from the Two objects dataset

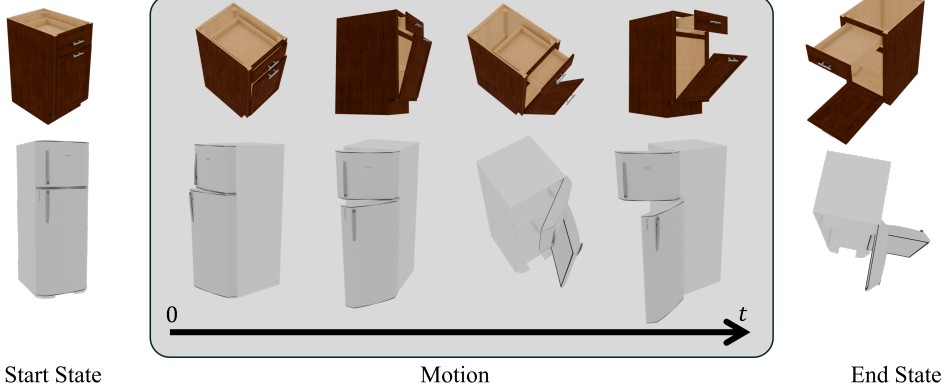

Start State    Motion    End State

Figure A2: Examples from the Three objects dataset

Table A1: Motion type and motion records across ten scenes on the Two-part objects dataset

| Scene | Blade 103706 | Fridge 10905 | Oven 101917 | Scissor 11100 | Stapler 103111 | Storage 45135 | USB 100109 | Washer 103776 |
|---|---|---|---|---|---|---|---|---|
| Motion type | Translate | Rotate | Rotate | Rotate | Rotate | Translate | Rotate | Rotate |
| Motion | $0 \rightarrow 0.5$ | $-110° \rightarrow 0°$ | $0° \rightarrow 90°$ | $45° \rightarrow -45°$ | $0° \rightarrow -80°$ | $0 \rightarrow 0.5$ | $0° \rightarrow -90°$ | $0° \rightarrow -60°$ |

Table A2: Motion types and ranges extracted from the two scenes of the Three-part objects dataset.

| Scene | Part ID | Motion Type | Range | Part ID | Motion Type | Range |
|---|---|---|---|---|---|---|
| Fridge 11304 | 0 | Rotate | $0 \rightarrow -180°$ | 1 | Rotate | $0° \rightarrow -90°$ |
| Storage 47024 | 0 | Rotate | $0° \rightarrow 90°$ | 1 | Translate | $0 \rightarrow 0.7$ |

Table A3: Motion types and ranges extracted from two additional scenes of the Complex objects dataset.

| Scene | Part ID | Motion Type | Range | Part ID | Motion Type | Range |
|---|---|---|---|---|---|---|
| Storage 47648 | 0 | Rotate | $0 \rightarrow 120°$ | 1 | Rotate | $0 \rightarrow -120°$ |
| | 2 | Rotate | $0 \rightarrow -60°$ | 3 | Rotate | $0 \rightarrow 60°$ |
| | 4 | Translate | $0 \rightarrow 0.1$ | 5 | Translate | $0 \rightarrow 0.16$ |
| Table 31249 | 0 | Translate | $0 \rightarrow 0.38$ | 1 | Translate | $0.35 \rightarrow 0$ |
| | 3 | Rotate | $0 \rightarrow -90°$ | 4 | Rotate | $0 \rightarrow 90°$ |

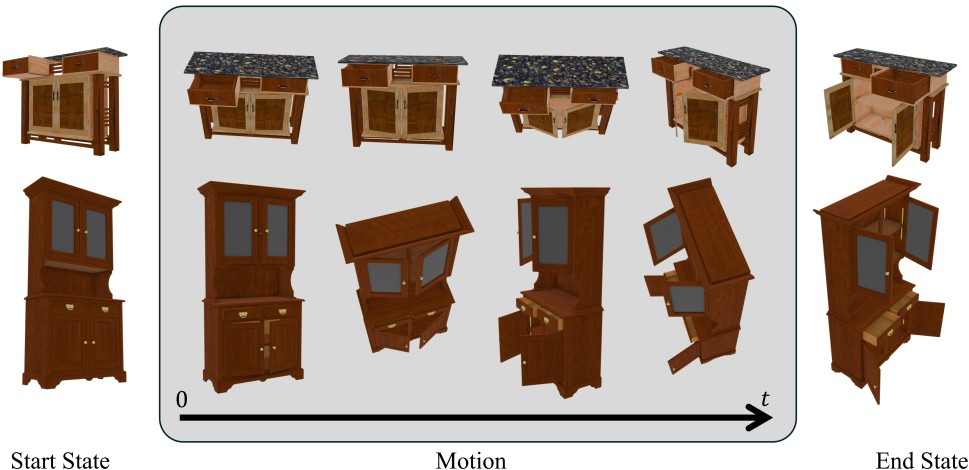

Start State           Motion           End State

Figure A3: Examples from the Complex objects dataset

Table A4: Comparison results with Video2Articulation (Peng et al., 2025). We report four metrics, *i.e.* Axis Ang (°), Axis Pose (0.1m), CD-m (mm) and CD-s (mm). Especially, for two-part objects, we report the metrics as mean$_{\pm std}$ across 10 trials. For a three-part object, we report the mean value on different moving parts. **Fail** represents that Video2Articulation fails to detect the correct part segmentation masks, and (%) shows the failure rate. **Bold** means better performance.

| | Method | Two-part | | | | Three-part | |
|---|---|---|---|---|---|---|---|
| | | Fridge | Storage | USB | Washer | Fridge (Joint0) | Fridge (Joint1) |
| Axis Ang (°) | Video2Articulation | $3.80_{\pm 0.00}$ | $6.53_{\pm 0.00}$ | $1.89_{\pm 0.00}$ | Fail (100%) | 2.17 | 1.35 |
| | Ours | $\mathbf{2.70}_{\pm 1.73}$ | $\mathbf{1.52}_{\pm 0.88}$ | $\mathbf{0.59}_{\pm 0.30}$ | $\mathbf{1.63}_{\pm 0.90}$ | **1.67** | **0.68** |
| Axis Pose (0.1m) | Video2Articulation | $0.95_{\pm 0.00}$ | — | $\mathbf{0.12}_{\pm 0.00}$ | Fail (100%) | 0.71 | **1.92** |
| | Ours | $\mathbf{0.86}_{\pm 0.34}$ | — | $1.45_{\pm 0.71}$ | $\mathbf{1.12}_{\pm 0.29}$ | **0.68** | 3.58 |
| CD-m (mm) | Video2Articulation | $8.06_{\pm 0.16}$ | $141.95_{\pm 12.02}$ | $24.68_{\pm 0.49}$ | Fail (100%) | 2.88 | 41.38 |
| | Ours | $\mathbf{2.21}_{\pm 0.18}$ | $\mathbf{18.95}_{\pm 2.57}$ | $\mathbf{0.89}_{\pm 0.10}$ | $\mathbf{21.03}_{\pm 1.02}$ | **2.12** | **3.85** |
| CD-s (mm) | Video2Articulation | $7.21_{\pm 0.12}$ | $8.66_{\pm 0.47}$ | $101.42_{\pm 0.72}$ | Fail (100%) | 44.45 | 44.45 |
| | Ours | $\mathbf{3.45}_{\pm 0.09}$ | $\mathbf{7.09}_{\pm 0.49}$ | $\mathbf{1.54}_{\pm 0.14}$ | $\mathbf{9.25}_{\pm 0.99}$ | **8.16** | **8.16** |

# D  ADDITIONAL RESULTS

## D.1  ADDITIONAL RESULTS ON OUR DATASET

As shown in Fig. A4, we provide more rendering results of our AIM. Additionally, we provide more visual comparisons among DTA, ArtGS, and ours. As shown in Fig. A5 and Fig. A6, we compare the rendering quality with ArtGS, and compare the part segmentation performance with DTA and ArtGS. Especially, all the results are with the start state and generated with the estimated motion parameters. From the results, we can see that our method achieves more stable and accurate part mobility analysis. Furthermore, the point clouds in Fig. A6 are presented to show the geometry recovery.

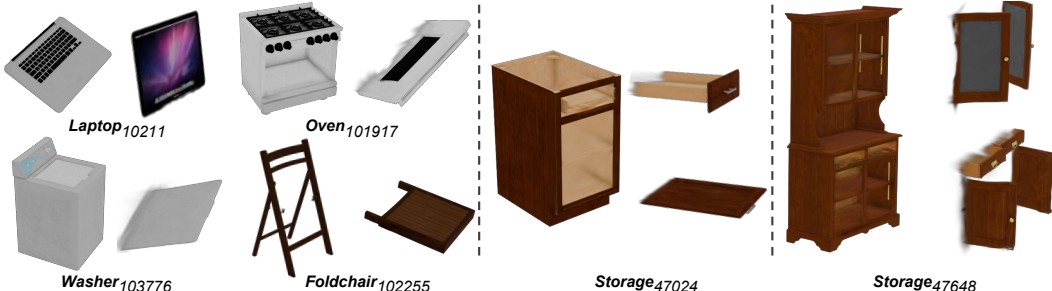

Figure A4: Rendering of our dual-Gaussian representation (**left**: static result $\{\mathcal{R}^S\}$, **right**: moving result $\{\mathcal{R}^{M,t=1}\}$). All objects (*category$_{instance}$*) are from PartNet-Mobility dataset (Mo et al., 2019).

## D.2  ADDITIONAL COMPARISON WITH PRE-TRAINED SEGMENTATION DRIVEN METHOD

As introduced in Sec. 2, our work primarily focuses on self-contained methods, namely approaches that can independently perform part-level mobility analysis without relying on any externally pre-trained segmentation models or segmentation-mask pools. Therefore, we select PARIS, DTA, and ArtGS as our baselines. Furthermore, motivated by the inherent limitations shared by these two-state-based methods, we propose AIM, which leverages motion cues from common close-to-open interaction videos to achieve part prior-free mobility analysis without any structural priors.

To further demonstrate the effectiveness of our method beyond the self-contained setting, we additionally report both quantitative and qualitative comparisons against the recent pre-trained segmentation-driven approach, Video2Articulation Peng et al. (2025). Since Video2Articulation requires preprocessing through Monst3r Zhang et al. (2024) and AutoSeg-SAM Zrporz (2024), we directly use the overlapping subset of objects provided in their released dataset. Specifically, we evaluate on four two-part objects (Fridge-10905, Storage-45135, USB-100109, Washer-103776) and one three-part object (Fridge-11304) and reproduce the official codes using the official settings. These results could be found in Tab. A4 and Fig. A7.

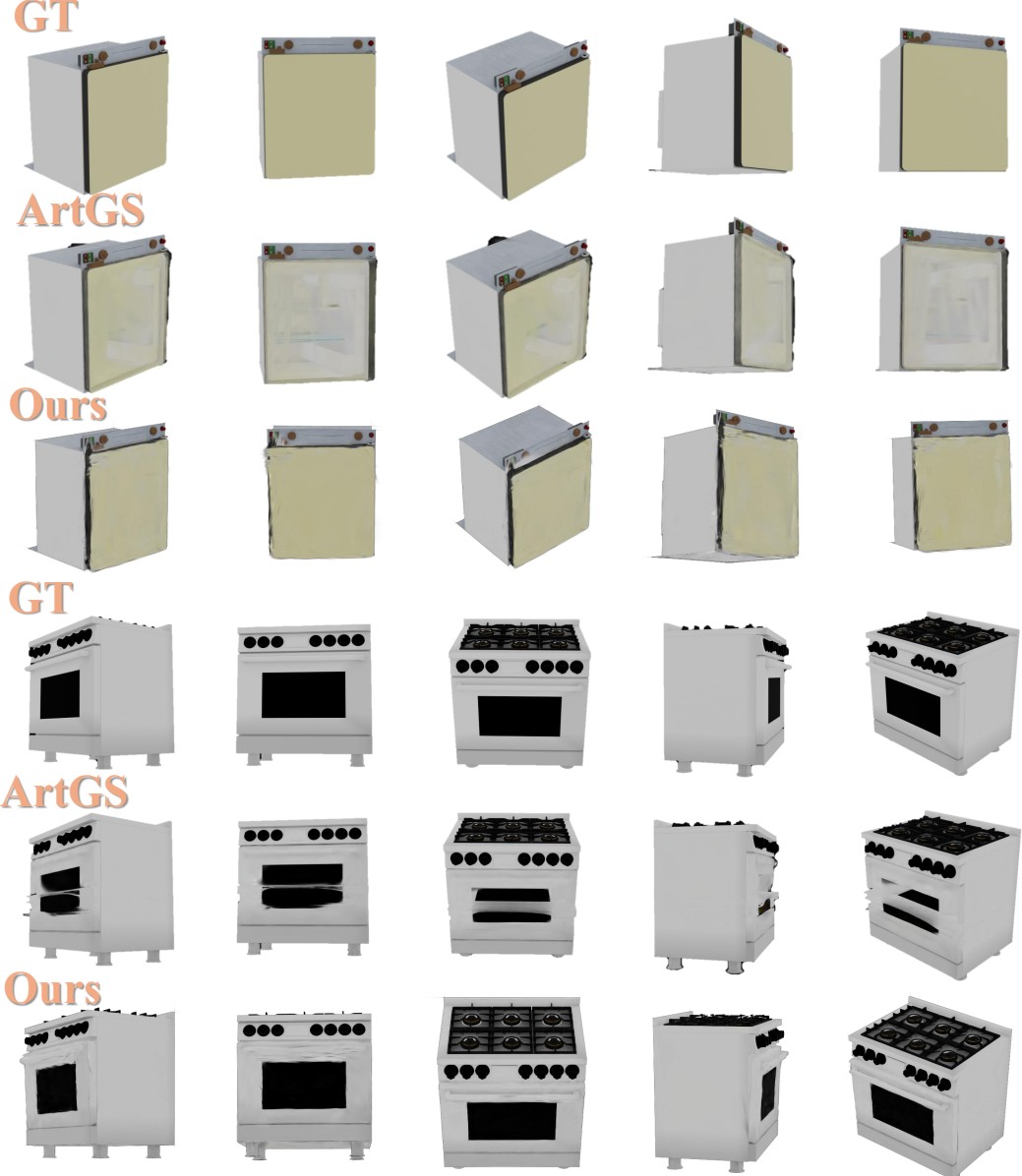

Figure A5: Rendering results based on articulation estimation parameters for the start state of two-part objects: **Top** (fridge): From the visualisation, it can be observed that the newly seen interior content severely influences the performance of ArtGS's articulation estimation, causing the door located inside the body. **Bottom** (oven): Similarly, during opening the oven, the newly seen content could not be well aligned between the two states, leading to wrong axis estimation and joint type estimation. The handle moves into the oven.

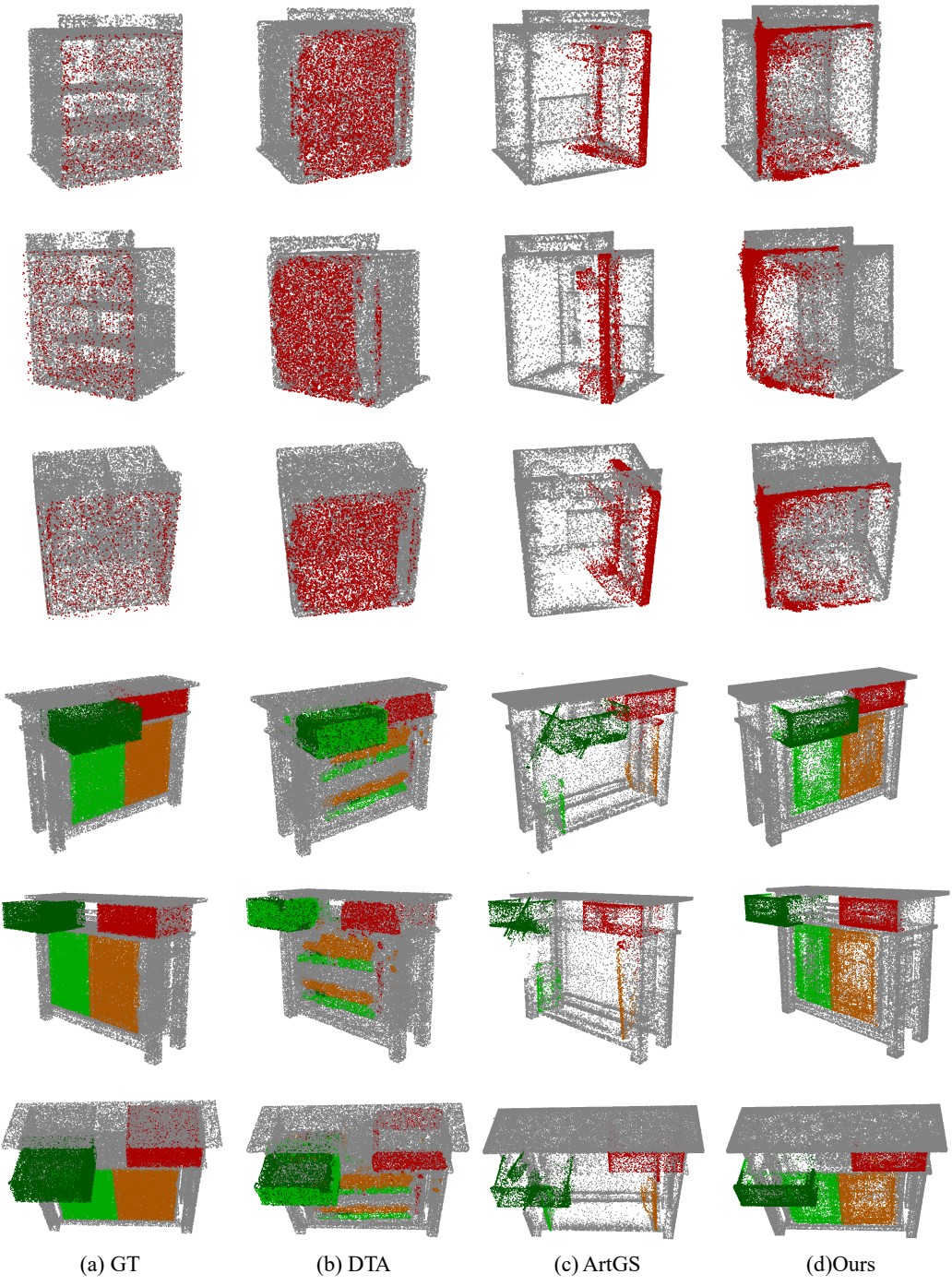

(a) GT          (b) DTA          (c) ArtGS          (d)Ours

Figure A6: Qualitative comparison between DTA, ArtGS and ours, w.r.t. GT.

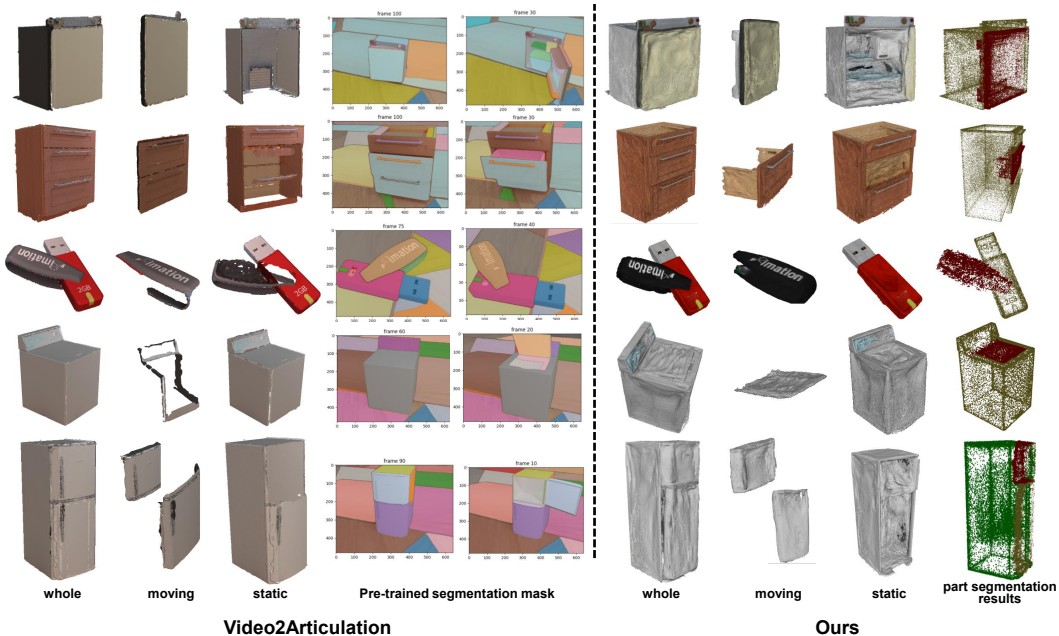

whole    moving    static    Pre-trained segmentation mask    whole    moving    static    part segmentation results

**Video2Articulation**    **Ours**

Figure A7: Visual results of Video2Articulation Peng et al. (2025) (Left) and ours (Right) (Notably, as the expensive pre-processing process of Video2Articulation, we directly use their released dataset to reproduce Video2Articulation, there is some colour difference). **Left**: From the visualisations, we observe that the pre-trained-segmentation–driven Video2Articulation method fails to predict correct moving parts when the underlying pre-trained segmentation models cannot provide reliable masks. Typical failure cases include mis-segmenting the drawer and its cabinet, or losing track of the inside of the refrigerator door once it opens. **Right:** In contrast, our approach performs part mobility analysis by directly exploiting motion cues from the interaction video. After achieving clean static–dynamic separation, we apply multi-model fitting based on the inferred motion patterns, which leads to consistently more robust and accurate results.

Table A5: **Quantitative evaluation of articulation estimation under the open-start and open-end conditions.** (a) Two-part; (b) Three-part; (c) Complex objects. For complex objects, we report the average of all moving parts. Due to the different magnitudes of part motion for revolute and prismatic joints, we report both of them. $-$ indicates prismatic joints w/o rotation axis.

(a) Two-part objects

| Metric | Method | Two-part objects | | | | | | |
|---|---|---|---|---|---|---|---|---|
| | | Fridge | Oven | Scissor | USB | Washer | Blade | Storage |
| Axis Ang | DTA | $0.08_{\pm 0.03}$ | $0.10_{\pm 0.04}$ | $0.05_{\pm 0.02}$ | $0.83_{\pm 0.49}$ | $2.05_{\pm 1.20}$ | $0.41_{\pm 0.11}$ | $0.14_{\pm 0.07}$ |
| | ArtGS | $0.00_{\pm 0.00}$ | $0.01_{\pm 0.00}$ | $0.07_{\pm 0.02}$ | $0.01_{\pm 0.00}$ | $0.03_{\pm 0.02}$ | $0.02_{\pm 0.00}$ | $0.00_{\pm 0.00}$ |
| | Ours | $0.19_{\pm 0.08}$ | $0.06_{\pm 0.03}$ | $0.21_{\pm 0.01}$ | $0.20_{\pm 0.06}$ | $0.05_{\pm 0.02}$ | $0.03_{\pm 0.01}$ | $0.05_{\pm 0.02}$ |
| Axis Pos | DTA | $0.01_{\pm 0.00}$ | $0.06_{\pm 0.03}$ | $0.01_{\pm 0.00}$ | $0.03_{\pm 0.02}$ | $3.05_{\pm 4.31}$ | $-$ | $-$ |
| | ArtGS | $0.00_{\pm 0.00}$ | $0.01_{\pm 0.00}$ | $0.00_{\pm 0.00}$ | $0.00_{\pm 0.00}$ | $0.00_{\pm 0.00}$ | $-$ | $-$ |
| | Ours | $0.04_{\pm 0.02}$ | $0.05_{\pm 0.04}$ | $0.00_{\pm 0.00}$ | $0.02_{\pm 0.01}$ | $0.02_{\pm 0.01}$ | $-$ | $-$ |
| Part Motion | DTA | $0.12_{\pm 0.04}$ | $0.20_{\pm 0.09}$ | $0.04_{\pm 0.02}$ | $0.66_{\pm 0.38}$ | $12.13_{\pm 11.07}$ | $0.00_{\pm 0.00}$ | $0.00_{\pm 0.00}$ |
| | ArtGS | $0.01_{\pm 0.00}$ | $0.04_{\pm 0.00}$ | $0.05_{\pm 0.00}$ | $0.03_{\pm 0.00}$ | $0.03_{\pm 0.00}$ | $0.00_{\pm 0.00}$ | $0.00_{\pm 0.00}$ |
| | Ours | $0.79_{\pm 0.34}$ | $0.46_{\pm 0.16}$ | $0.58_{\pm 0.05}$ | $1.19_{\pm 0.07}$ | $0.10_{\pm 0.04}$ | $0.00_{\pm 0.00}$ | $0.00_{\pm 0.00}$ |

(b) Three-part objects

| | Methods | $D_0$ | | | $D_1$ | | |
|---|---|---|---|---|---|---|---|
| | | Axis Ang | Axis Pos | Part Motion | Axis Ang | Axis Pos | Part Motion |
| Storage 47254 | DTA | 0.09 | 0.02 | 0.07 | 0.32 | $-$ | 0.00 |
| | ArtGS | 0.04 | 0.00 | 0.01 | 0.05 | $-$ | 0.00 |
| | Ours | 0.18 | 0.05 | 0.22 | 0.05 | $-$ | 0.00 |
| Fridge 10489 | DTA | 0.26 | 0.01 | 0.19 | 0.18 | 0.01 | 0.26 |
| | ArtGS | 0.02 | 0.00 | 0.01 | 0.00 | 0.00 | 0.05 |
| | Ours | 0.09 | 0.01 | 0.42 | 0.06 | 0.04 | 0.85 |

(c) Complex objects

| | $\downarrow$ | $\text{Ang}_{avg}$ | $\text{Pose}_{avg}$ | $\text{Motion}^r_{avg}$ | $\text{Motion}^p_{avg}$ |
|---|---|---|---|---|---|
| Storage 47648 | ArtGS | 10.18 | 0.43 | 10.91 | 0.13 |
| | Ours | 0.08 | 0.24 | 1.62 | 0.03 |
| Table 31249 | ArtGS | 0.02 | 0.00 | 0.01 | 0.00 |
| | Ours | 0.36 | 0.00 | 0.27 | 0.00 |

Table A6: **Mesh reconstruction comparison under the open-start and open-end condition.** (a) Two-part objects; (b) Three-part objects; (c) Complex objects. For two-part objects, we report CD distance (mm) as mean$\pm_{std}$ across 5 trials. For three-part and complex objects, we only report the mean value, while we report the average CD for movable parts. Lower ($\downarrow$) is better.

(a) Two-part objects

| Metric | Method | Two-part objects | | | | | | |
|---|---|---|---|---|---|---|---|---|
| | | Fridge | Oven | Scissor | USB | Washer | Blade | Storage |
| CD-S | DTA | $0.62_{\pm0.02}$ | $4.59_{\pm0.13}$ | $0.71_{\pm0.51}$ | $3.19_{\pm1.07}$ | $1.69_{\pm1.10}$ | $0.80_{\pm0.10}$ | $2.78_{\pm0.04}$ |
| | ArtGS | $0.50_{\pm0.00}$ | $4.74_{\pm0.02}$ | $0.82_{\pm0.23}$ | $2.58_{\pm0.01}$ | $0.96_{\pm0.01}$ | $0.71_{\pm0.00}$ | $4.65_{\pm0.03}$ |
| | Ours | $0.53_{\pm0.00}$ | $4.59_{\pm0.18}$ | $0.57_{\pm0.00}$ | $2.95_{\pm0.13}$ | $0.85_{\pm0.01}$ | $0.72_{\pm0.00}$ | $5.84_{\pm1.60}$ |
| CD-m | DTA | $0.30_{\pm0.01}$ | $0.47_{\pm0.01}$ | $0.46_{\pm0.13}$ | $3.52_{\pm1.92}$ | $1.38_{\pm0.65}$ | $3.28_{\pm0.33}$ | $0.40_{\pm0.00}$ |
| | ArtGS | $0.27_{\pm0.00}$ | $0.52_{\pm0.00}$ | $0.79_{\pm0.25}$ | $2.27_{\pm0.05}$ | $0.27_{\pm0.01}$ | $2.80_{\pm0.14}$ | $1.61_{\pm0.02}$ |
| | Ours | $0.25_{\pm0.00}$ | $0.63_{\pm0.06}$ | $0.49_{\pm0.00}$ | $1.33_{\pm0.04}$ | $0.32_{\pm0.01}$ | $0.75_{\pm0.01}$ | $2.94_{\pm0.31}$ |

(b) Three-part objects

| | $\downarrow$ | DTA | ArtGS | Ours |
|---|---|---|---|---|
| Storage | CD-s | 1.01 | 0.95 | 1.58 |
| *47254* | CD-$D_0$ | 0.49 | 0.25 | 0.18 |
| | CD-$D_1$ | 1.11 | 0.41 | 0.46 |
| Fridge | CD-s | 2.66 | 1.97 | 2.12 |
| *10289* | CD-$D_0$ | 3.56 | 1.26 | 1.26 |
| | CD-$D_1$ | 2.78 | 0.76 | 0.69 |

(c) Complex objects

| | $\downarrow$ | ArtGS | Ours |
|---|---|---|---|
| Storage$_{47648}$ | CD-s | 1.52 | 1.64 |
| | CD-m$_{avg}$ | 3.89 | 4.36 |
| Table$_{31249}$ | CD-s | 2.11 | 2.08 |
| | CD-m$_{avg}$ | 3.60 | 4.19 |

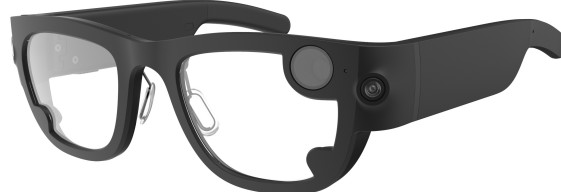

Figure A8: The Meta Project Aria Glasses.

## D.3 ADDITIONAL RESULTS UNDER THE OPEN-START AND OPEN-END SETTING

As shown in Tab. A5 and Tab. A6, we evaluate all methods under the same input setting used in ArtGS and DTA, namely, the open-start and open-end configuration. Meanwhile, we render the sequences using the motion parameters provided by ArtGS. Under this setting, both ArtGS and DTA achieve good articulation estimation, particularly because the geometric correspondence between the two states is clear and the part number is known. This also makes the articulation estimation of ArtGS stable, with almost zero variance in the evaluation of articulation estimation. Meanwhile, without structural priors as input, our AIM also achieves accurate and stable articulation estimation through dual-Gaussian–based dynamic–static disentanglement and robust motion-based Sequential RANSAC. As shown in Tab. A5, Tab. A6, our results are comparable to these optimisation-based baselines, and in challenging complex-object cases, AIM is more stable. Meanwhile, AIM's part segmentation, driven only by motion cues in interaction videos and without requiring the number of parts, remains on par with or even surpasses recent state-of-the-art ArtGS.

## D.4 ADDITIONAL RESULTS ON REAL-WORLD DATA

### D.4.1 REAL-WORLD DATASET ACQUISITION

To better support natural human–object interaction during data capture, we leverage the Meta Project Aria Glasses (as shown in Fig. A8) to record the interaction video. In detail, videos are recorded in real time using the device's built-in fisheye cameras while the user observes the target object and manipulates its movable parts. As shown in the video, during the interaction process, the user first walks into the scene and observes both the surrounding environment and the articulated objects in their closed-start state. To achieve the automatic pipeline, the user then signals the beginning of interaction by using the hand to touch the target object ( *i.e.* the oven). While manipulating the movable part, the user freely moves the head to observe the object from different viewpoints. The

hand is then removed to inspect the object again. The user subsequently touches another object ( *i.e.* the storage) and repeats the same manipulation-and-observation procedure.

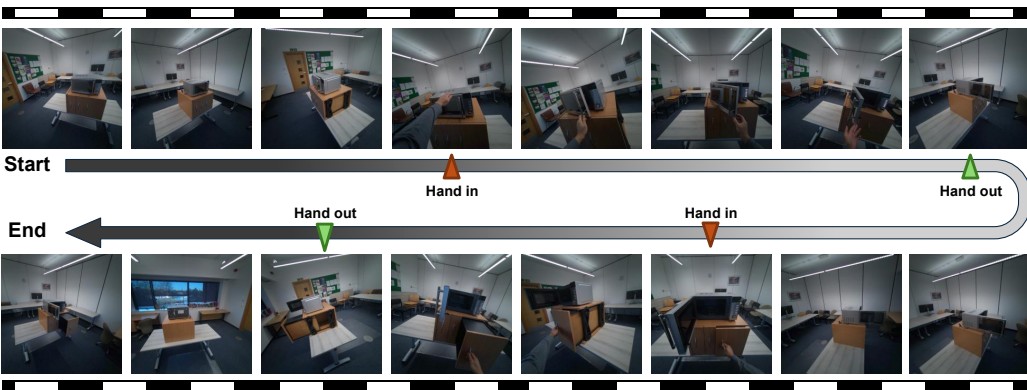

Figure A9: The captured interaction videos (The video could be found in the Supplementary Material). In particular, we use the hand as an indicator to automatically detect the motion start (*hand in*) and end times (*hand out*), enabling a fully automated data-processing pipeline.

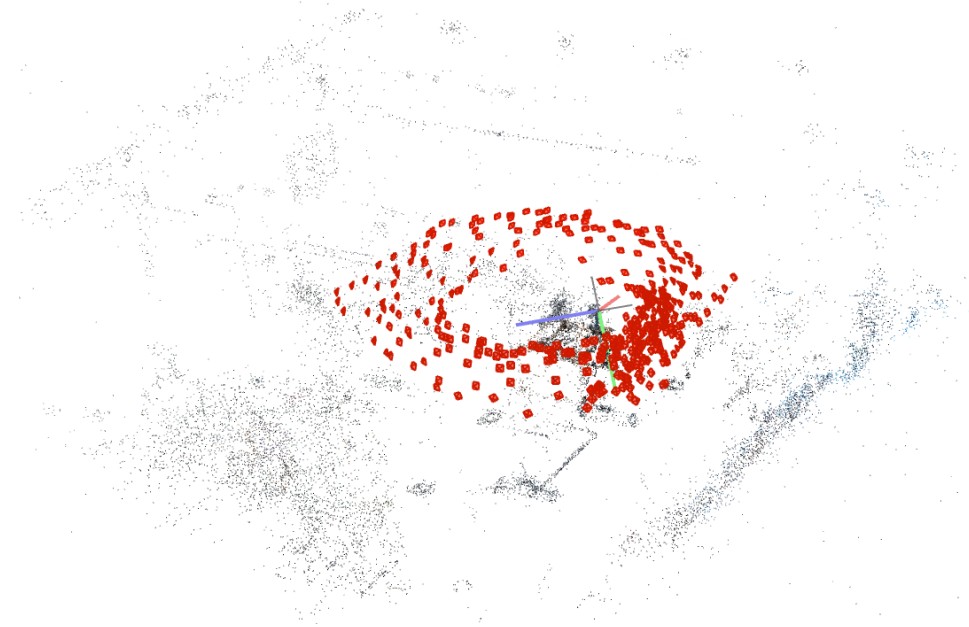

Figure A10: The estimated pose of real-world video via using COLMAP Schonberger & Frahm (2016).

For data processing, we first extract the recordings from the Aria glasses. Since the RGB cameras on Aria glasses are fisheye cameras, we apply the official Project Aria toolkit (Engel et al., 2023) to rectify each frame and re-project it into a pinhole camera mode. This produces a set of undistorted pinhole frames along with their corresponding timestamps (see Fig. A9). We then extract keyframes with FFMPEG and manually filter the frames to remove blurred or low-quality images. Finally, as shown in Fig. A10, the curated images are fed into COLMAP (Schonberger & Frahm, 2016) to estimate camera poses via structure-from-motion, which are used as inputs for subsequent reconstruction. Especially, we follow the process, introduced in video2articulation Peng et al. (2025), to obtain the whole articulated object (*i.e.* oven and storage) via Grounded SAM2 Ravi et al. (2024); Ren et al. (2024) with the two text prompts: "silver microwave" and "wooden storage". The final inputs to AIM consist of ***100 start-state frames and 87 motion frames*** for the oven sequence, and

***77 start-state frames and 58 motion frames*** for the storage sequence. Although the number of motion frames is much smaller than in our rendered dataset, the following results show that AIM still performs robustly and accurately under this limited motion input.

### D.4.2   EXPERIMENTAL RESULTS ON REAL-WORLD DATA

As shown in Fig. A11 and Fig. A12, we present our results on the real-world captured sequences. AIM performs robust and accurate part mobility analysis purely from RGB inputs, without any structural prior knowledge. Notably, AIM reliably predicts the correct joint-axis direction and achieves low-error articulation estimation, such as the oven door's nearly 85° opening motion, which is predicted as about 82°. Moreover, our SDMD module correctly reassigns newly revealed static regions during motion, such as the oven interior. For the storage example, despite significant occlusion from the oven and the user's hand, AIM still produces correct part-level segmentation and articulation estimation. These results further confirm the strong motion analysis ability of dual-Gaussian representation and the generalisation capability of AIM in challenging real-world scenarios.

**Limitations**:  From the real-world data, we observe that when motion introduces structural ambiguities—*particularly those caused by specular reflections, such as the glass door of the oven in the video*—our vanilla 3DGS-based reconstruction in AIM can be affected. In future work, we plan to further address such challenges, for example, by incorporating depth information to improve robustness under complex lighting and reflective surfaces. Moreover, as discussed in Sec. 5, we will extend AIM to more diverse and larger real-world scenes in future work.

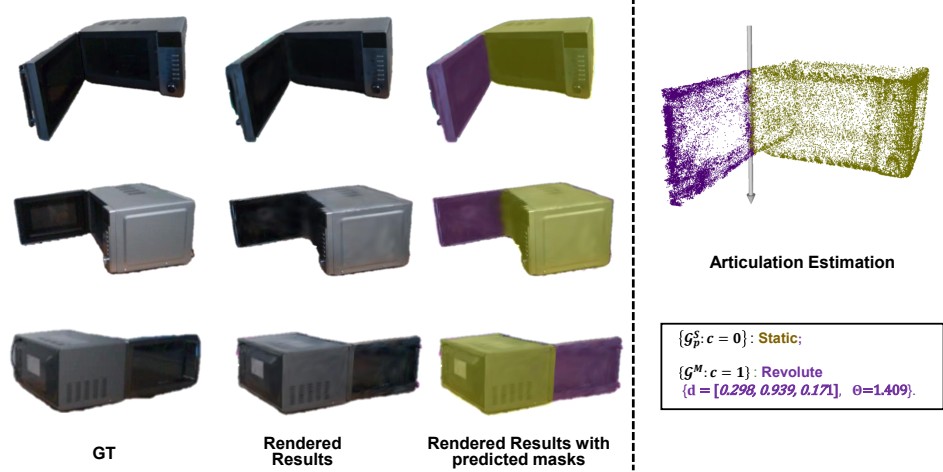

Figure A11: Qualitative results of our AIM on the real-world data of the oven. **Left:** Comparison between the ground-truth views and rendered views. ***Besides, we provide the rendered masks based on our dual-Gaussian representation (via directly changing the spherical harmonics of Gaussians)***. Due to the strong specular reflections on the oven's glass door, the appearance of the moving part undergoes frequent and significant changes during interaction. Despite this challenge, our dual-Gaussian representation still achieves clean dynamic–static disentanglement by relying on stable motion cues. Moreover, the SDMD module reliably reassigns the newly revealed static interior regions back to the static base as the motion unfolds, further improving the quality of disentanglement and reconstruction. **Right:** Our prior-free part mobility analysis. Based on our dual-Gaussian representation, we can easily infer the trajectories of moving Gaussians, and obtain the articulation parameters based on the optimisation-free and robust sequential RANSAC without any prior structural knowledge.

## E   ABLATION STUDY

In Fig. A13, we present qualitative ablations by removing individual modules, highlighting the importance of each component. As shown in Fig. A14, we also provide the comparisons between our dual Gaussian representation and the deformable Gaussian. The dynamic-static disentanglement of our dual Gaussian representation can better track the moving regions and support more accurate trajectory-based part segmentation.

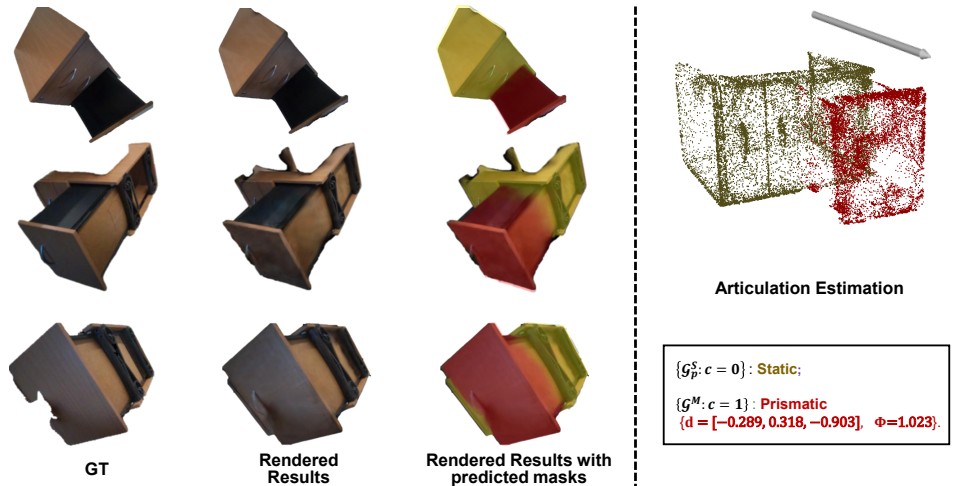

Figure A12: Qualitative results of our AIM on the real-world data of storage. **Left:** Comparison between the ground-truth views and rendered views. Besides, we provide the rendered masks based on our dual-Gaussian representation (via directly changing the spherical harmonics of Gaussians). Although large regions of the storage are occluded by the oven and the hand during the interaction video (see Fig. A9), AIM remains robust and accurately performs dynamic–static disentanglement, enabling clean separation of the static base and the moving part purely from motion cues. **Right:** Our prior-free part mobility analysis. Based on our dual-Gaussian representation, we can easily infer the trajectories of moving Gaussians, and obtain the articulation parameters based on the optimisation-free and robust sequential RANSAC without any prior structural knowledge. (Notably, for prismatic joint, we do not consider the axis pose).

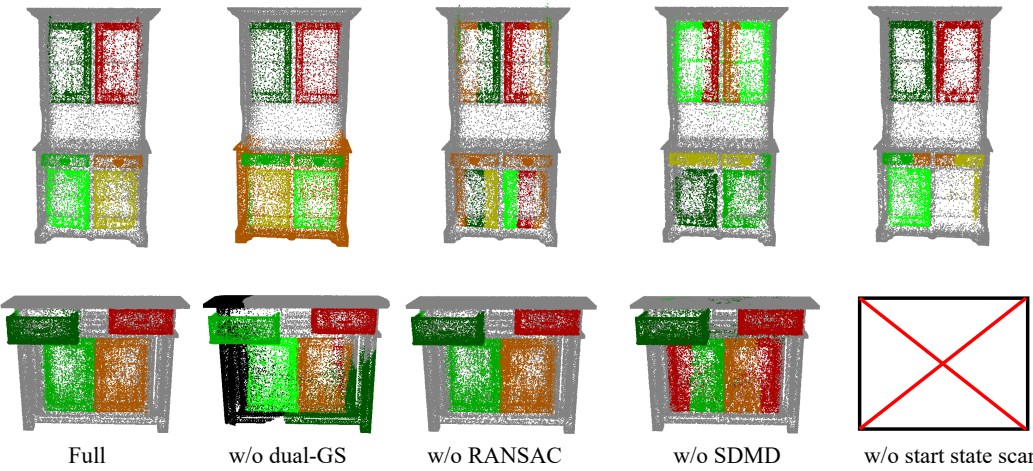

Figure A13: Qualitative comparisons for the ablation studies.

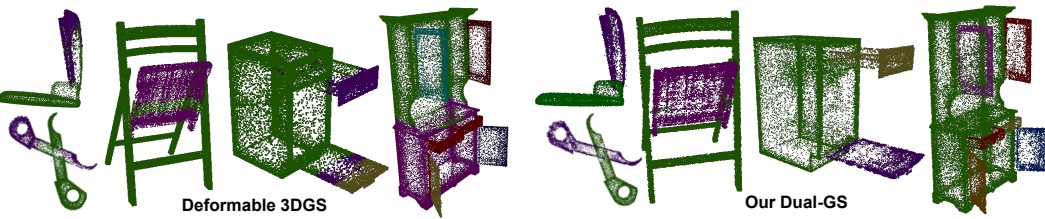

Figure A14: Part segmentation results based on Deformable 3DGS and our dual-Gaussian representation. We cluster Gaussians using the same sequential RANSAC settings; colours denote groups. Static noise in Deformable 3DGS noticeably degrades the segmentation.

