# OpenReview forum: "Articulation in Motion: Prior-free Part Mobility Analysis for Articulated Objects By Dynamic-Static Disentanglement"
_ICLR.cc/2026/Conference — ICLR 2026 Poster_

### Official Review · Reviewer_3jJi · 2025-10-17

**Soundness:** 3
**Presentation:** 3
**Contribution:** 2
**Rating:** 4
**Confidence:** 4

**Summary:**

This paper proposes Articulation in Motion (AIM), which reconstructs articulated objects from motion videos instead of two-state scans. It uses a dual-Gaussian representation (static + deformable) to separate moving/static parts, then applies Sequential RANSAC on motion trajectories to segment parts and estimate joint parameters without knowing part count.

**Strengths:**

1. Novel Problem Formulation: The shift from two-state (start-end) analysis to continuous motion video is well-motivated and more practical. The paper correctly identifies that many articulated objects cannot be well-represented by just two discrete states, especially when interior parts are gradually revealed.

2. Technical Contribution - Dual-Gaussian Representation and Static-During-Motion Detection (SDMD): The proposed dual-Gaussian approach ({G^S} for static, {G^M,t} for moving) addresses a real limitation of prior work. The joint optimization strategy with selective pruning is useful.

3. Prior-Free Framework: Eliminating the need to know the number of parts is a significant practical advantage. The sequential RANSAC approach for automatic part discovery is well-suited to this problem.

**Weaknesses:**

1. Unrealistic Trajectory Assumption (T=2 Sampling): The paper's core technical assumption is critically flawed for real-world videos. The method samples T=2 timesteps: {0 → 0.5, 0 → 1}, which assumes that all the parts of an articulated object should have different state at t=0, 0.5, 1.0. However, in fact, when we capture videos of interacting with objects in real scenes, we usually can only interact with different parts one by one. This means that in a video with hundreds of frames, only a few dozen frames of a certain part may be in motion.  For example:
- t=0.0: Door closed (0°)
- t=0.5: Door closed (0°)      ← No motion detected
- t=0.7: Door opens and then closes
- t=1.0: Door closed (0°)      ← No motion detected!

2. Questionable Baseline Results: The reported performance of ArtGS and DTA is suspiciously poor and inconsistent with the original paper and other current works [1, 2, 3]. For example, Table 2 shows ArtGS achieving 473.72mm CD-m on Blade and 155.65mm on Washer—these are catastrophically bad results. Similarly, Table 3 reports axis angular errors of 89.91° and 89.92° on Fridge and Oven, meaning the estimated joint axes are essentially perpendicular to ground truth, suggesting complete method failure rather than mere performance degradation. These results are orders of magnitude worse than what ArtGS reports in their original paper and what subsequent works[1, 2, 3] have achieved when reproducing ArtGS. If the method truly failed this catastrophically, the authors should explain why, but no such analysis is provided. Other recent papers successfully reproduce ArtGS with reasonable performance, making these results highly suspicious. Additionally, the performance of ArtGS on complex objects is even better than that on two-part objects. This abnormal phenomenon is also an unreasonable reproduction resultWithout credible baseline implementations, the claimed improvements cannot be validated—this is a fundamental flaw that invalidates the entire experimental section.

3. Lack of results in real-world scenario: despite emphasizing practical advantages over two-state methods, the paper provides zero real-world experiments. The authors never demonstrate that their system works on actual captured videos of people opening doors or manipulating objects. Without real-world validation, it's unclear whether the method's restrictive assumptions (fixed T={0,0.5,1.0} timesteps) actually provide any practical advantage over existing approaches.

[1] Self-Supervised Multi-Part Articulated Objects Modeling via Deformable Gaussian Splatting and Progressive Primitive Segmentation
[2] Part2GS: Part-aware Modeling of Articulated Objects using 3D Gaussian Splatting
[3] REArtGS: Reconstructing and Generating Articulated Objects via 3D Gaussian Splatting with Geometric and Motion Constraints

**Questions:**

See weakness.

---

> ### Author Response · Authors · 2025-11-22
> **Response to Reviewer 3jJi**
>
> We thank the reviewer for the thoughtful and thorough comments. We appreciate the reviewer’s recognition of our key strength, namely the value of moving beyond two-state analysis to continuous video, the dual-Gaussian representation, and the prior-free sequential RANSAC for part discovery. We also appreciate that the reviewer engaged deeply with the paper, and we respond to each of the concerns raised below.
> ### Response to weaknesses
> **W1: Clarification on trajectory assumption.**
>
> - **About the input**: As stated in the Abstract and Introduction, our method is inspired by prior two-state approaches. However, to overcome their limitations--*namely, the reliance on structural priors and the need for consistent cross-state geometric correspondence*--we instead aim to achieve prior-free part mobility analysis purely from continuous motion cues in human–object interaction videos. In real scenarios, such interactions naturally follow a *closed-start → open-end* progression; therefore, we normalize the continuous motion timeline by setting t = 0 to the closed-start frame and t = 1 to the open-end frame, which is analogous to the two distinct motion states used in prior two-state methods.
>
> - **About the sampling**: Regarding the two time windows 0→0.5 and 0→1, we would like to clarify that they are **only used for support-set selection**. We compute the inlier errors under multiple time windows to improve the robustness of RANSAC support selection. Now we further clarify this point in Sec 3.2 of the revised version (highlighted in blue).
>
> We also fully acknowledge the reviewer’s point that, in real scenes, interaction could involve sequential part manipulations, or may contain repetitive or highly nonlinear motions. While this is indeed a meaningful scenario, it is beyond the scope of the problem addressed in this work. Nevertheless, we are actively exploring extensions such as: 1) improving the time-dependent deformation field beyond the current D-3DGS formulation (e.g., more expressive time-conditioned dynamic Gaussians), 2) incorporating sliding-window motion detection for sequential or repeated interactions, and 3) allowing light human-guided cues (e.g., briefly observing the environment after opening a door) to help confirm whether motion occurs in a given temporal window. These are promising directions for future work.
>
> We would like to emphasize that this work explores the potential of combining classical multi-model fitting with modern 3D Gaussian representations. Compared with prior two-state methods that rely on structural priors and consistent cross-state geometric correspondence, AiM achieves accurate articulated-object part mobility analysis, *whether for single-part or multi-part motion*, purely from motion cues and without any structural prior knowledge.

---

> ### Author Response · Authors · 2025-11-22
>
> **W2: Clarification on baseline performance**
>
> Although the performance of ArtGS and DTA may seem unexpected at first glance, we would like to clarify that the results are correct and fully consistent with the inherent assumptions of these methods. We detail the reasons below.
>
> - **I. All baseline reproductions strictly follow the official papers and their released code.**
>     All baseline methods (ArtGS, DTA, PARIS) were reproduced strictly following:
>
>     - the hyperparameters reported in their papers,
>     - the official implementations released by the authors
>     - the same training protocol described in their Method sections.
>
>     **The only difference** lies in the input data, not in the reproduction process.
>     Our data follows a *closed-start → open-end* interaction sequence.
>     In contrast, all existing two-state methods—including ArtGS, DTA, Part2GS, and REArtGS, use open-start → open-end data where all interior structures must be already visible at the start to make sure the consistent geometric correspondence.
>
>     We have documented the full dataset details in Appendix. A (Table A1–A3 and Fig. A1–A3). Although differences in Blender version, lighting, and camera settings prevent pixel-identical replication, all objects are rendered with high fidelity, ensuring that these differences do not affect the experimental conclusions.
>
> - **II. Two-state methods fail under closed-start → open-end is structural rather than a reproduction flaw.**
>
>     We respectfully but firmly disagree with the claim that our results represent “an unreasonable reproduction.” However, the large drop in performance of ArtGS/DTA is fully aligned with their algorithmic assumptions and explicitly stated limitations. ArtGS depends critically on canonical state initialization, obtained by computing Chamfer distances between start-state and end-state point clouds to infer a pseudo mid-state.
>     This initialization is used to derive:
>     - canonical → start deformation
>     - canonical → end deformation
>     - part centers and blend-skinning weights
>
>     ArtGS explicitly states in its Limitations: “As demonstrated in Sec. 5.3, faulty initialization of $G^c$ and C can lead to significant performance degradation, particularly for complex objects with multiple movable parts.” This is exactly the situation in our closed-start → open-end data, i.e., fridge, washer, and oven reveals interior only at end-state. Because the interior structures do not exist in the start-state point cloud, the Chamfer-based canonical initialization in ArtGS becomes corrupted. This leads to predictable failure modes: unseen interiors are labeled as dynamic, true dynamic parts are labeled as static, and the joint type is mispredicted—producing large axis-angle errors (often near 90°) and large part-motion errors. Since we follow the official ArtGS evaluation code, a wrong joint-type prediction (e.g., classifying a revolute joint as a primitive joint) directly manifests as these large errors. To better describe it, we mark such cases as wrong type detecction (WT) in the revised version and add visualizations (Fig.6 in the revised version) to clearly illustrate these failure modes.
>
>     In total, their peformance is not due to incorrect reproduction, but arise naturally from the inherent limitations of two-state methods. Because these methods rely on consistent geometric correspondence between the two states, their assumptions break down in our closed-start → open-end setting, where previously unseen interior structures become visible only at the end. This realistic setting exposes the fundamental limitation of prior approaches and further motivates the design of AiM.
>
> - **III. Why multi-part objects appear “better” for ArtGS in our results.** When more dynamic parts are present and the part number is given, the “newly revealed” interior content contributes proportionally less to the canonical misalignment, thus reducing the damage caused by visibility asymmetry. We add the visualization reuslts on Fig.6 to better show the performance.
>
> - **IV. Additional clarification and new experiments.**
> To further verify that our reproductions are correct and make it clear, in the revised submission: 1) We added the visualizations and analysis in Sec 4.2 (highlighed in blue) to clearly show how ArtGS fails. 2) We additionally conduct the experiments, following the original open-start → open-end setting provided by ArtGS/DTA. The results could be found in Appendix C.3.
>
> To further support reproducibility and clarity, and facilitate future research in the community, we will release all the data, rendering codes, and pipeline codes.

---

> > ### Author Response · Authors · 2025-11-22
> >
> > **W3: About real-world examples.**
> >
> > We thank the reviewer for raising this point. In this work, we use the rendered PartNet-Mobility data primarily to enable quantitative evaluation, as it provides precise ground-truth annotations that are unavailable in existing monocular real-world articulated datasets. As noted in Sec. 5, extending AiM to real-world captured videos is valuable for verifying its generalization capability.
> > ***We are currently running experiments on real-world RGB interaction videos and start-state multi-view scans. Due to the tight rebuttal timeline, processing is still ongoing. We will update the real-world results during the rebuttal period as soon as they become available.***
> >
> > We thank reviewer 3jJi once again for the constructive feedback. We hope that our detailed responses and revisions adequately address the raised concerns.

---

> > > ### Comment · Reviewer_3jJi · 2025-11-28
> > >
> > > Thank you for the detailed response to my questions, which solved some of my concerns. Although I still think the trajectory assumption will limit the application scenarios of this method, if it can be verified using real data, I will consider improving my score.

---

> > > > ### Author Response · Authors · 2025-12-01
> > > >
> > > > We appreciate the reviewer’s follow-up comment and are glad that our previous responses the reviewer's concerns. We have now updated the **real-world experiments** in *Appendix. C.4* in the revised version (highlighted in blue), where we describe our real-world data acquisition pipeline, provide implementation details, and include corresponding visual results. *Our real-world experiments clearly validate that the trajectory assumption made in AiM is reasonable and holds in practical interaction scenarios*. Moreover, despite the limited number of motion frames and the more challenging real-world conditions, the proposed AiM continues to produce accurate and stable part mobility analysis, thanks to its motion-based dual-Gaussian representation and robust, prior-free sequential RANSAC. As discussed in Sec. 5 of the main paper, we plan to further extend AiM to larger and more complex real-world environments in future work.

---

### Official Review · Reviewer_ZaoU · 2025-10-28

**Soundness:** 3
**Presentation:** 2
**Contribution:** 2
**Rating:** 4
**Confidence:** 4

**Summary:**

The paper tackles the task of reconstructing articulated objects from a user-object interaction video and a start-state scan. Previous methods mainly focus on reconstruction from two different articulation states, which the authors think limits their applications and performance. The authors propose a dual-Gaussian scene representation learned from an initial 3DGS scan. It then applies RANSAC on motion trajectories to segment parts and estimate joint parameters.

**Strengths:**

1. The paper explores a new setting that copes with start-state scans and a human-object interaction video instead of two-state multi-view observations in previous papers. This sounds more reasonable and practical than previous papers.
2. The proposed method use RANSAC to perform part discovery, eliminating the necessity of knowing part numbers ahead.
3. The idea of dual-Gaussian representation that differentiate static and moving Gaussians sound reasonable to me.

**Weaknesses:**

1. One of the claim of this paper is that it copes with a human-object interaction video and a start-state scan instead of two-state multi-view observations. But the dataset used in the paper seems to be rendered from PartNet-Mobility. Some real-world examples should be helpful.
2. The baseline choices seem weird. The chosen baselines are mainly PARIS, DTA, and ArtGS, which mainly focuses on two-state observations. Some more plausible baseline may be Video2Articulation[1].
3. The authors should also elaborate on how they perform these baselines as the setting are very much different.

[1] Generalizable Articulated Object Reconstruction from Casually Captured RGBD Videos

I slightly lean towards borderline reject, and I will change my rating if my concerns are addressed and consider other reviewers' comments.

**Questions:**

See above

---

> ### Author Response · Authors · 2025-11-22
> **Response to Reviewer ZaoU**
>
> We thank the reviewer for the thoughtful comments, and for recognizing the reasonableness of our motion-based dual-Gaussian representation, and the benefits of our prior-free RANSAC-based part mobility analysis. We address the concerns below.
>
>
> **Clarification on the summary:**
> There might be a minor misunderstanding in the reviewer’s summary that we would like to clarify. Our dual-Gaussian representation is not learned from the initial 3DGS scan alone. Instead, we leverage the joint optimization of static 3DGS and moving deformable 3DGS to analyze the motion cues in the video and achieve dynamic-static disentanglement. The initial 3DGS model (start-state 3DGS) is guided to specialize in representing the static base, while the moving Gaussian set tracks all moving parts revealed in the video. We hope this clarification will make our contributions clearer.
>
> ### Response to Weaknesses
> **W1: About real-world examples**
>
> Thanks for your valuable suggestion. In this work, we utilized the PartNet-Mobility rendered data primarily to provide a quantitative evaluation benchmark, as it offers precise ground truth, which is typically unavailable in existing monocular real-world articulated object datasets.
>
>
> - ***Adding the experiments on the real-world interaction videos***: As discussed in the future work of Sec.5 in main paper, we fully agree that providing the real-world demonstration can better show the generalization capability. ***We are currently conducting real-world experiments based on captured RGB interaction videos and start-state multi-view scans. Due to the tight rebuttal timeline, the processing of these sequences is still ongoing. We will update the results here shortly as soon as they are ready.***

---

> > ### Author Response · Authors · 2025-11-22
> >
> > **W2: About the baseline choices**
> >
> > Thanks for the reviewer's suggestion and careful review. Our selection of PARIS, DTA, and ArtGS is motivated by the fact that these methods are the closest to our research problem, since they all aim to perform part mobility analysis of articulated objects, including part segmentation and articulation parameter estimation, and interactive reconstruction. In contrast, the approaches, such as Video2Articulation, RSRD, and POD, depend heavily on pre-trained part segmentation masks generated by large-scale segmentation models (e.g., SAM2 [1] in Video2Articulation, or DINOv2 [2] + SAM [3] in RSRD). As discussed in our Related Work (Sec. 2), this reliance implies that if the pre-trained model fails to segment an articulated part correctly, these methods cannot recover the true part structure or articulation. This is fundamentally different from the motivation of our work: part segmentation is the critical first step for part mobility analysis, whereas these video-based methods bypass this step by relying on externally provided segmentation masks rather than discovering parts from motion cues.
> >
> > **Moreover, as suggested, we further added both qualitative and quantitative comparisons with Video2Articulation** in revised Appendix C.2 (highlighted in blue). Since Video2Articulation requires preprocessing through monst3r and AutoSeg-SAM, we directly use the overlapping subset of objects provided in their released dataset. Specifically, we evaluate on four two-part objects (Fridge-10905, Storage-45135, USB-100109, Washer-103776) and one three-part object (Fridge-11304) and reproduce the official codes using the official settings.
> >
> > We run each method ten times, report the mean and variance for two-parts object, and report the average performance for three-part object. Notably, since the part segmentation of Video2Articulation depends on AutoSeg-SAM, segmentation failures might occur, as shown in their original paper. For cases where the method fails in all ten runs, we report them directly as "Fail". In addition, we include the failure rate across the ten runs. The comparison is reported in the table below. As shown in the experimental results, on most metrics, our method consistently outperforms Video2Articulation. In particular, for the Washer object, the pre-trained segmentation model fails to continuously produce a mask for the washer's lid during the video, causing Video2Articulation to fail in all ten runs. For more comparison results, we kindly refer the reviewer to the qualitative results provided in Appendix C.2 of the revised version.
> >
> >
> >
> > | Metric | Method | Fridge-10905 | Storage-45135 | USB-100109 | Washer-103776 | Fridge-11304 (Joint0) | Fridge-11304 (Joint1) |
> > |--------|---------|--------------|----------------|------------|----------------|-------------------------|-------------------------|
> > | **Axis Ang** | Video2Articulation | 3.80±0.00 | 6.53±0.00 | 1.89±0.00 | Fail (100%) | 2.17 | 1.35 |
> > | **Axis Ang** | **Ours** | **2.70±1.73** | **1.52±0.88** | **0.59±0.30** | **1.63±0.90** | **1.67** | **0.68** |
> > | **Axis Pose (0.1m)** | Video2Articulation | 0.95±0.00 | ——— | 0.12±0.00 | Fail (100%) | 0.71 | **1.92** |
> > | **Axis Pose (0.1m)** | **Ours** | **0.86±0.34** | ——— | 1.45±0.71 | **1.12±0.29** | **0.68** | 3.58 |
> > | **CD-m (mm)** | Video2Articulation | 8.06±0.16 | 141.95±12.02 | 24.68±0.49 | Fail (100%) | 2.88 | 41.38 |
> > | **CD-m (mm)** | **Ours** | **2.21±0.18** | **18.95±2.57** | **0.89±0.10** | **21.03±1.02** | **2.12** | **3.85** |
> > | **CD-s (mm)** | Video2Articulation | 7.21±0.12 | 8.66±0.47 | 101.42±0.72 | Fail (100%) | 44.45 | 44.45 |
> > | **CD-s (mm)** | **Ours** | **3.45±0.09** | **7.09±0.49** | **1.54±0.14** | **9.25±0.99** | **8.16** | **8.16** |
> >
> > [1] Ravi, Nikhila, et al. "Sam 2: Segment anything in images and videos." arXiv preprint arXiv:2408.00714 (2024).
> >
> > [2] Oquab, Maxime, et al. "Dinov2: Learning robust visual features without supervision." arXiv preprint arXiv:2304.07193 (2023).
> >
> > [3] Kirillov, Alexander, et al. "Segment anything." Proceedings of the IEEE/CVF international conference on computer vision. 2023.

---

> ### Author Response · Authors · 2025-11-22
>
> **W3: How baselines are executed in our setting**
>
> We appreciate the reviewer’s suggestion. For experiments, our differences from existing methods mainly lie in the following aspects:
>
> - **Differences in rendering Data**. Existing methods only provide start-state and end-state multi-view images. Thus, for fair comparison, we rendered our own dataset on the same 3D models, following exactly the same training data protocol described in their papers—specifically, randomly sampling 100 upper-hemisphere views for both states (Sec. 4.1). However, details of the rendering pipeline, *e.g.*, Blender version, render engine, illumination intensity/distance, and camera configurations, are not specified in prior works. Therefore, our rendered images are not pixel-identical to theirs, although the view sampling strategy is fully consistent.
>
> - **Differences in input conditions** Prior works typically assume *open-start → open-end* inputs, where both states clearly reveal all parts and thus exhibit strong geometric correspondence. In contrast, our setting follows the more common *closed-start → open-end* interaction, where interior parts become visible only during motion. This reflects how articulated objects are normally used, rather than imposing additional difficulty. The motion types, magnitudes, and rendering examples are detailed in Appendix B.1 for full transparency.
>
> We would like to sincerely thank prior works for their contributions and for releasing high-quality code, which greatly facilitated our reproduction. We confirm that all baseline models were trained and evaluated on our rendered dataset strictly following the settings described in their papers and official implementations.

---

### Official Review · Reviewer_Rvjo · 2025-10-31

**Soundness:** 3
**Presentation:** 3
**Contribution:** 2
**Rating:** 6
**Confidence:** 3

**Summary:**

The paper introduces a novel framework called ”Articulation in Motion” (AIM), which performs reconstruction, segmentation, and articulation analysis of articulated objects. The authors propose a practical solution that requires only a user-object interaction video and a start-state scan to generate a part-level decomposition, an abstraction of the articulation kinematics, and a 3D digital replica. The framework is developed over three stages: the first consists of a 3D Gaussian splatting reconstruction of the object in an initial static state; in the second step, the static 3D Gaussian splatting is optimised, and jointly another 3D Gaussian splatting representation of the moving parts is generated from the video. At this stage, the novel static parts revealed by the motion of the articulated parts are detected using a static-during-motion (SDMD) module and added to the static representation. In the third stage, with a sequential RANSAC, the trajectories of each moving primitive are grouped to perform motion-based part segmentation. The proposed solution has been tested over a dataset provided by the authors, and it has been compared with state-of-the-art methods by overcoming them, especially on the
segmentation of the articulated parts.

**Strengths:**

• The internal organisation of the paper is a significant strength, ensuring that each component of the proposed solution is introduced and explained coherently and sequentially.
• The proposed framework introduces a strong methodological contribution, integrating several state-of-the-art algorithms in a novel manner to solve an important problem, such as the analysis and reconstruction of articulated objects without any prior knowledge of the object's moving parts.
• The quantitative and qualitative evaluations reported in the experiment section confirm the validity of the proposed solution. In particular, the proposed method outperforms all state-of-the-art methods across metrics related to part segmentation, while achieving above-average results in the other metrics.

**Weaknesses:**

- The dual Gaussian representation is introduced as novel, but it seems the same as the dense static Gaussians and sparse dynamic Gaussians introduced in ArtGS.
- Apart from the novelty of the video interaction, it is not quite clear where the novelty here is w.r.t. ArtGS.

**Questions:**

1. Given the interaction video, is it possible that two parts are in motion at the same time? Or is the interaction constrained to a single specific part motion?
2.  The number of states is not clear.
3.  How is the freezing and unfreezing of the static Gaussian managed?
4. I would appreciate some clarification on the dual optimisation.

---

> ### Author Response · Authors · 2025-11-22
> **Response to Reviewer Rvjo**
>
> We thank the reviewer for the strong recognition of our work, especially for acknowledging our methodological contribution, clear organization, and superior performance on part mobility analysis for articulated objects. We address these concerns below:
>
> ### **Response to weaknesses**
> **W1: The novelty of dual-Gaussian representation.**
>
> We would like to highlight that our Dual-Gaussian representation is fundamentally different in function and principle from the dense static and sparse dynamic Gaussians in ArtGS.
>
> - The core of ArtGS is to optimize part-center positions and per-Gaussian assignment probabilities, and to estimate articulation parameters by learning per-Gaussian transformations from an initialized canonical (mid-state) Gaussian representation to the two given states. The static–dynamic detection is only an auxiliary step used to refine the canonical Gaussian initialization. Concretely, ArtGS compares Gaussians between the two states and applies a manually set threshold: Gaussians with large spatial discrepancies are marked as dynamic, and those with small discrepancies are treated as static. These thresholded “static” Gaussians are then merged into the canonical Gaussian set.
>
> - Our dual-Gaussian representation learns dynamic–static disentanglement directly from the motion cues in the interaction video, rather than relying on manually defined thresholds for hard separation. To this end, we propose the joint optimization strategy to optimize a static Gaussian set and a deformable Gaussian set.
>
>
> **W2: Novelty w.r.t. ArtGS.**
>
> We thank the reviewer for the valuable feedback. We would like to clarify that beyond the use of video input, our key contribution lies in ***how to extract motion cues in interaction videos***, and ***how to leverage the captured motion information to achieve the part mobility analysis without part-level structural priors***. Concretely, I) based on the motion cues, we propose a novel dual-Gaussian representation to reliably anchor the static base and track moving Gaussians through joint optimization, providing a clean dynamic–static disentanglement. II) Built on the dual-Gaussian representation, we subtly use a training-free, robust sequential RANSAC to cluster motion trajectories from the moving Gaussians. This method is simple, effective, and does not rely on any part-level structural priors, automatically inferring both the number of parts and their articulation parameters purely from motion cues.

---

> ### Author Response · Authors · 2025-11-22
>
> ### **Response to Questions**
> **Q1: About the motion condition in the video**
>
> For our method, the motion condition in the video is not limited. Since we apply Sequential RANSAC to group consistent motion patterns for part segmentation, our method naturally supports multiple motions occurring simultaneously in the video. As described in Sec. 4.1, our multi-part experiments are also configured such that multiple parts move at the same time.
>
>
> **Q2: The number of states.**
>
> The number of "states" in AiM simply corresponds to the number of video frames used during dual-Gaussian training. As introduced in Sec.4.1, we typically use 200 frames (i.e., 200 motion states), while we use 500 frames for multi-part objects. Since AiM operates on continuous interaction videos, the number of motion states is freely controllable through the sampling frequency of the input frames.
>
> **Q3. How is the freezing and unfreezing of the static Gaussian managed?**
>
> Freezing and unfreezing is indeed a key part of our joint optimization.
> As described in Sec. 3.1 and Fig. 4, we first reconstruct the start-state Gaussian set $\{G^S\}$  from multi-view scans. At this stage, $\{G^S\}$ contains both the static base and the future moving parts. Our goal is to let $\{G^S\}$ specialize to the static base, while a separate deformable 3DGS $\{G^M\}$ captures the dynamic components purely from motion cues in the video. Therefore, after obtaining $\{G^S\}$, we initialize a new moving Gaussian set $\{G^M\}$ following D-3DGS[3], and train an MLP to predict time-dependent shifts in center position and rotation. During optimization, we jointly render $\{G^S\}$∪$\{G^M, t\}$ and supervise the rendering output with the photometric loss using the ground-truth images captured at the timestep $t$.
>
> To achieve clean dynamic–static disentanglement, we freeze all attributes of $\{G^S\}$—including center, rotation, scale, and SH coefficients---**except opacity** during the first 10k iterations. Meanwhile, all attributes of $\{G^M\}$ are fully optimized. Under this setup, the combination of the two Gaussian sets must explain the video frames; thus, the opacity of any dynamic component inside $\{G^S\}$ naturally decays toward zero, because the deformable Gaussian set $\{G^M, t\}$ more accurately fits those moving regions. Then, the opacity-based pruning will remove these dynamic components inside $\{G^S\}$ with the near-zero opacity values, yielding the static-base Gaussians $\{G_p^S\}$.
> After 10k iterations, we unfreeze all attributes of $\{G_p^S\}$ and jointly optimize it together with $\{G^M\}$ to model the full object. Aside from this freezing/unfreezing schedule, the entire optimization strictly follows standard 3DGS and Deformable 3DGS settings. The choice of 10k iterations is not a fixed hyperparameter—it simply aligns with the default densify_until_iter parameter (15k) of 3DGS[4] and D-3DGS. We revised Sec. 3.1 to make it more clear (highlighted in blue).

---

> ### Author Response · Authors · 2025-11-22
>
> ### **Response to Questions**
> **Q4: Clarification on the dual optimisation.**
>
> As shown in Fig.4 in main paper, our joint optimization consists of two stage.
>
> - Stage 1 (first 10k iterations): As introduced in the response to Q3, this stage aims to remove dynamic components from the start-state Gaussian set {$\{G^S\}$}. To achieve this, we freeze all attributes of {$\{G^S\}$}--**except opacity**--and keep the moving Gaussian set {$\{G^M, t\}$} **fully trainable**. We then jointly render {$\{G^S\}$}∪{$\{G^M, t\}$} and supervise the result with a photometric loss, which updates the opacity of {$\{G^S\}$}, all attributes of {$\{G^M, t\}$}, and the time-dependent deformation field (as in D-3DGS). During this stage, we apply adaptive density control (densification + pruning) to {$\{G^M\}$}, while {$\{G^S\}$} undergoes opacity-based pruning. As a consequence, the opacity of dynamic regions in {$\{G^S\}$} naturally converges to near-zero and these gaussians are pruned, leaving only the Gaussians for static base, namely {$\{G_p^S\}$}. Meanwhile, {$\{G^M, t\}$} captures and tracks all dynamic components, including any newly revealed content that is not visible at the start (e.g., the interior of a fridge when the door opens).
>
> - Stage 2 (from 10k iteration to the end of training): During this stage, we first unfreeze all attributes of {$\{G_p^S\}$} and continue to jointly render and optimize {$\{G_p^S\}$}∪{$\{G^M, t\}$} with the same photometric supervision. Before the densify_until_iter iteration, we apply standard adaptive density control to both {$\{G_p^S\}$} and {$\{G^M\}$}  (densification + pruning, as in vanilla 3DGS[4] / D-3DGS). All Gaussian attributes of {$\{G_p^S\}$} and {$\{G^M\}$}, as well as the deformation field of {$\{G^M\}$}, are jointly optimized. Importantly, between 10k iteration and densify_until_iter iteration, we also run static-during-motion detection module every 2k iterations before densification/pruning, in order to reassign newly revealed but static regions from the moving set back into {$\{G_p^S\}$} (see Sec. 3.1 for details). We will release the complete implementation to illustrate the dual-Gaussian joint optimization process and to facilitate future research.
>
> We have revised the description in Method (highlighed in blue) to make this clearer.
>
>
> [1]Wu, Di, et al. "Reartgs: Reconstructing and generating articulated objects via 3d gaussian splatting with geometric and motion constraints." arXiv preprint arXiv:2503.06677 (2025).
>
> [2]Guo, Junfu, et al. "Articulatedgs: Self-supervised digital twin modeling of articulated objects using 3d gaussian splatting." Proceedings of the Computer Vision and Pattern Recognition Conference. 2025.
>
> [3]Yang, Ziyi, et al. "Deformable 3d gaussians for high-fidelity monocular dynamic scene reconstruction." Proceedings of the IEEE/CVF conference on computer vision and pattern recognition. 2024.
>
> [4]Kerbl, Bernhard, et al. "3D Gaussian splatting for real-time radiance field rendering." ACM Trans. Graph. 42.4 (2023): 139-1.

---

> > ### Comment · Reviewer_Rvjo · 2025-11-24
> > **answer to authors**
> >
> > Thank you for the detailed answers to my questions, which have clarified my doubts. I'm looking forward to the "experiments on real-world RGB interaction videos and start-state multi-view scans." If there is time to complete these experiments, in response to reviewer 3iji.

---

> > > ### Author Response · Authors · 2025-12-01
> > >
> > > We appreciate the reviewer’s follow-up comment and are glad that our previous responses clarified the reviewer's doubts. We have now updated the **real-world experiments** in *Appendix. C.4* in the revised version (highlighted in blue), where we describe our real-world data acquisition pipeline, provide implementation details, and include corresponding visual results. Despite the limited number of motion frames and the more challenging real-world conditions, the proposed AiM continues to produce accurate and stable part mobility analysis, thanks to its motion-based dual-Gaussian representation and robust, prior-free sequential RANSAC. As discussed in Sec. 5 of the main paper, we plan to further extend AiM to larger and more complex real-world environments in future work.

---

### Official Review · Reviewer_ch7L · 2025-11-02

**Soundness:** 4
**Presentation:** 3
**Contribution:** 4
**Rating:** 8
**Confidence:** 3

**Summary:**

Paper addresses the problem of piecewise-rigid reconstruction via combining old-school computer vision (sensible priors in a dual Gaussian formulation and sequential RANSAC) with modern Gaussian-based 3d rendering.

This is a solid piece of engineering that shows that the use of established CV methods can substantially contribute to the performance of modern systems.

+ Comfortably state-of-the-art
+ Good ablation studies showing the contribution of each component.
+ Thoughtful qualitative results/figures making it clear how each component is useful.

It's worth mentioning that this paper opens the door to more sophisticated forms of model fitting than sequential RANSAC. However, the proposed benchmark is saturated by this approach. More challenging data would be needed before anything better than greedy optimization is needed.

**Strengths:**

This is an extremely well-motivated approach where the design decisions have a clear and obvious contribution and that substantially improves over multiple recent existing approaches.

The writing is clear, and the design decisions are well supported by experimental evaluation.

+ Comfortably state-of-the-art
+ Good ablation studies showing the contribution of each component.
+ Thoughtful qualitative results/figures making it clear how each component is useful.

This paper opens the door to more sophisticated forms of model fitting than sequential RANSAC. However, the proposed benchmark is saturated by this approach. More challenging data would be needed before anything better than greedy optimization is needed.

**Weaknesses:**

The clarity of the approach probably works against this paper (at least at review time, this clarity is a benefit when published), and I suspect that some of the other reviews may complain about lack of novelty as it looks obvious in hindsight. Of course, if it was obvious, the existing approaches would already be doing something similar, and the performance improvement would be less noticeable.

I think describing the dual Gaussian approach as prior free is somewhat misleading. The reason it works well is because it injects a healthy prior that most of the world is static, which makes the actual task much easier.

The dataset is small by modern ML dataset standards, consisting of few enough sequences that they can be named and discussed individually. However, this is fairly common in 3d reconstruction, where if you want high-quality ground truth, you typically have to pay the cost in terms of a small number of evaluation sequences.

**Questions:**

-

---

> ### Public Comment · ~Rango_Barry1 · 2025-11-18
> **clearly sus**
>
> No Question

---

> ### Author Response · Authors · 2025-11-22
> **Response to Reviewer ch7L**
>
> We sincerely thank the reviewer for the positive and encouraging feedback on our work. We deeply appreciate your recognition of our motivation and finding our work to be a solid piece of engineering. We address your informed weakness as follows:
>
> ### Response to Weaknesses
> **W1: For the clarity.**
>
> We appreciate the reviewer’s recognition of the clarity of our approach, and we view this as a positive reflection of the principled and well-motivated design of our method. We further hope it inspires the community to explore this direction more deeply.
>
> **W2: "Prior-Free" Claim**
>
> Thanks for the suggestion. Our use of “prior-free” refers to removing part-level structural priors (e.g., the number of parts, joint type) for part-level mobility analysis, including part-level segmentation and per-part articulation parameter estimation. We do agree that the Dual-Gaussian representation leverages the relative motion between the static base and moving parts for dynamic–static separation. We have clarified it in the Abstract (highlighted in blue).
>
> **W3: For Dataset**
>
> We thank the reviewer for the constructive and positive feedback. As outlined in Sec. 5, in the future, we aim to construct a larger scene-level dataset and further investigate the integration of multi-model fitting with advanced 3D neural representations. In addition, we are currently conducting an additional experiment on *real-world captured interaction videos* to further demonstrate the generalization capability of our AiM. The results will be updated as the evaluation completes.

---

> > ### Comment · Reviewer_ch7L · 2025-11-25
> >
> > Thank you for the response.
> >
> > Looking forward to seeing an update if completed in time.

---

> > > ### Author Response · Authors · 2025-12-01
> > >
> > > We appreciate the reviewer’s follow-up comment. We have now updated the **real-world experiments** in *Appendix. C.4* in the revised version (highlighted in blue), where we describe our real-world data acquisition pipeline, provide implementation details, and include corresponding visual results. Despite the limited number of motion frames and the more challenging real-world conditions, the proposed AiM continues to produce accurate and stable part mobility analysis, thanks to its motion-based dual-Gaussian representation and robust, prior-free sequential RANSAC. As discussed in Sec. 5 of the main paper, we plan to further extend AiM to larger and more complex real-world environments in future work.

---

### Author Response · Authors · 2025-12-01
**General Response**

We sincerely thank all reviewers and ACs for their time, constructive feedback, and thoughtful recognition of the methodological contributions of our work. Following the reviewers’ comments, we have substantially revised the paper. All updates are highlighted in blue in the revised version. Below we summarize the major changes:

### Additional Experiments

1. *Added Real-world experiments  (ZaoU–W1, 3jJi–W1/W3, ch7L–W3: Appendix C.4).*
We added real-world experiments together with detailed descriptions of data acquisition, processing, and visualized our results. These results demonstrate that our trajectory assumption is reasonable in practice, and that AiM exhibits strong generalization and robustness even under limited and challenging real-world inputs.

2. *Added comparisons to pre-trained segmentation–based methods (ZaoU–W2: Appendix C.2).*
We added both qualitative and quantitative comparisons with recent SoTA pre-trained segmentation-based method. These comparisons further validate the soundness of our baseline choices and highlight AiM’s performance advantages.

3. *Additional experiments under the open-start → open-end setting (ZaoU–W3, 3jJi–W2: Appendix C.3, L468).*
To address concerns regarding the performance drop of prior two-state methods under the closed-start → open-end setting, we additionally evaluated all methods under the original open-start → open-end setting used in their papers. The reproduced results match the reported numbers, confirming correctness. Under this setting, AiM—*without any structural priors*—achieves comparable or even superior results, further demonstrating its robustness and generalization capability.

### Clarifications and Additional Details

1. *Clarified the definition of “prior-free” (ch7L–W2: L25).*
We refined the Abstract section to avoid over-claiming and to precisely define what structural priors AiM removes.

2. *Expanded technical introduction of previous methods (Rvjo–W1/W2: L70).*
We improved the Related Work section to clearly articulate the conceptual differences between AiM and existing two-state or segmentation-dependent approaches.

3. *Added more technical details of the pipeline (Rvjo–Q3/Q4: L220, L242, L303, L363).*
We added details about our dual-Gaussian optimization schedule, sequential RANSAC-based articulation estimation to improve clarity.

4. *Added citations and discussion on pre-trained segmentation–based video approaches (ZaoU–W2: L161).* We explicitly discussed pre-trained segmentation-based methods and clarified how our motivation differs from these approaches.

We thank the reviewers once again for their insightful feedback and the effort invested in evaluating our submission. Although the rebuttal phase does not allow further discussion, we are glad that our responses have resolved the concerns raised by the reviewers (e.g., Rvjo, 3jJi). We believe the revisions have significantly improved the paper, and the real-world results further verify the strength of our work.

---

### Meta-Review · Area_Chair_ZExp · 2026-01-01

**Summary:**

The paper initially received mixed reviews with scores of 4, 4, 6, and 8.

The main issue raised by most reviewers pertains to the experiments. In the revised version, the authors included additional experiments in the appendices to address these concerns. Additional clarifications and more technical details were provided to address other presentation-related issues.

Although real-world experiments reveal the limitations of the proposed method, the area chair agrees that the rebuttal and revision have adequately addressed the reviewers' main concerns. As a result, the area chair recommends accepting this paper.

**Reviewer Concerns:**

The authors conducted further real-world experiments as requested by reviewers. The results are promising, but they also reveal limitations when applied to real-world data with complex lighting and reflective surfaces. While the reviewers' primary concerns have been addressed, more thorough evaluations are needed in future work.

**Reviewer Scores:**

The area chair expects that  **Reviewer ch7L** would maintain a score of 8.

**Reviewer Rvjo** would either maintain a score of 6 or increase it to 8 as the authors provide *"experiments on real-world RGB interaction videos and start-state multi-view scans."*

Both **Reviewer ZaoU** and **Reviewer 3jJi** indicated that they would consider raising their scores if their concerns were addressed. Since the primary issues related to the experiments have been adequately addressed in the revision, which includes additional experimental results, the area chair expects that both reviewers will have a more positive opinion.

The final scores are expected to be **8**, **6**, **6**, and **6**.

---

### Decision · Program_Chairs · 2026-01-26

Accept (Poster)